# CENP-E initiates chromosome congression by opposing Aurora kinases to promote end-on attachments

Kruno Vukušić ✉ & Iva M. Tolić ✉

Accurate cell division relies on rapid chromosome congression. The kinetochore motor protein CENP-E/kinesin-7 is uniquely required for congression of polar chromosomes. It is currently assumed that CENP-E drives congression by gliding kinetochores along microtubules independently of their biorientation. Here, by studying chromosome movement under different levels of CENP-E activity, we favor an alternative model in which CENP-E initiates congression by promoting stabilization of end-on attachments. In this way, CENP-E accelerates congression initiation without significantly contributing to subsequent movement. Stabilization of end-on attachments on polar chromosomes without CENP-E is delayed due to Aurora kinase-mediated hyperphosphorylation of microtubule-binding proteins and expansion of the fibrous corona. CENP-E counters this by reducing Aurora B-mediated phosphorylation in a BubR1-dependent manner, thereby stabilizing initial end-on attachments, facilitating removal of the fibrous corona, and triggering biorientation-dependent chromosome movement. These findings support a unified model of chromosome movement in which congression is intrinsically coupled to biorientation.

To ensure equal segregation into two daughter cells, duplicated chromosomes achieve biorientation by attaching to microtubules extending from opposite spindle halves, thereby satisfying the spindle assembly checkpoint (SAC)[1–3]. During early mitosis, chromosomes move towards the equatorial region of the spindle in a process known as chromosome congression[4]. Chromosomes positioned at the center of the nucleus at the onset of mitosis rapidly achieve biorientation and are characterized by shorter congression movements, without requiring kinetochore molecular motors[4,5]. In contrast, chromosomes positioned closer to spindle poles, referred to as polar chromosomes, require the activity of the kinetochore motor protein CENtromere-associated Protein-E (CENP-E/kinesin-7) for more pronounced congression movements[4]. Recent studies identified that polar chromosomes are particularly prone to alignment failure during shortened mitosis in non-transformed cells and unperturbed mitosis in transformed cells[6–8]. Thus, the congression of polar chromosomes seems to exhibit distinct dynamics, regulatory mechanisms, and final efficiency[9].

CENP-E function in chromosome movement has been studied extensively[1,4,10–21]. The prevailing model posits that CENP-E drives the gliding of polar kinetochores, which are laterally attached to microtubules, toward the equatorial plate through its microtubule plus-end directed motor activity[22–25]. This model suggests that chromosome congression is decoupled from chromosome biorientation, which occurs near the metaphase plate[19,23]. The preference for which motor drives kinetochore movement in the tug-of-war between plus-end directed CENP-E and minus-end directed dynein is influenced by several factors. This includes a single-site phosphorylation of CENP-E at Threonine 422 (T422) by Aurora kinases near the spindle poles[26] and the increased detyrosination of tubulin on kinetochore fibers[27]. Both modifications enhance the motor activity of CENP-E[28,29]. CENP-E is recruited to an unattached outer kinetochore by BuBR1[30], and to a fibrous corona, which sheds from the kinetochore upon stable end-on attachment formation[15,29,31], by the Rod-ZW10-Zwilch (RZZ) and Dynein-Dynactin complexes[30,32].

Division of Molecular Biology, Ruđer Bošković Institute, Zagreb, Croatia. ✉e-mail: kvukusic@irb.hr; tolic@irb.hr

Despite the established requirement of CENP-E for chromosome congression, several observations cannot be explained by the current models. First, polar chromosomes begin to move toward the spindle center even in the absence of CENP-E activity, but they fail to properly align and move toward the nearest spindle pole[33], contradicting the gliding model of CENP-E function. Second, polar chromosome congression still occurs even in the complete absence of CENP-E, although it is markedly delayed[4,12,26,34,35]. The relationship between CENP-E-driven and CENP-E-independent congression remains unknown. This is underscored by a paradox in Aurora kinase function, as they appear to promote congression by activating CENP-E[26] while simultaneously suppressing end-on attachment formation, a process also reported to involve CENP-E[4,36]. As a result, the mechanisms underlying chromosome congression and the coordination of the key molecular players involved in this process remain poorly understood.

Here, we uncover that chromosome congression unfolds in two distinct biomechanical steps: initiation and movement. CENP-E is crucial only for initiating congression by facilitating the formation or stabilization of end-on attachments, rather than directly driving chromosome movement to the spindle equator, which occurs with the same dynamics regardless of CENP-E presence or activity. We arrived at this conclusion by employing a large-scale live-cell imaging strategy based on lattice light-sheet microscopy (LLSM), which enabled the simultaneous observation and tracking of numerous chromosome congression events across many cells in both space and time. By integrating this imaging approach with reversible chemical inhibition, protein depletion, and expression of phospho-mutants of key congression regulators, we delineated the hierarchical molecular networks that govern chromosome congression. Our data demonstrate that CENP-E initiates the stabilization of end-on attachments, a critical step in congression which precedes chromosome movement. The initiation of chromosome congression involves the formation and stabilization of end-on attachments, evidenced by super-resolution microscopy of tubulin and recruitment of the end-on attachment marker Astrin, together with a gradual reduction of SAC and fibrous corona proteins on kinetochores. Aurora kinases, as master inhibitors of congression initiation, oppose this process by destabilizing end-on attachments and promoting the expansion of the fibrous corona. CENP-E counteracts the activity of Aurora kinases in a BubR1-dependent manner, promoting rapid stabilization of end-on attachments and subsequent chromosome congression movement. In summary, we demonstrate that chromosome biorientation is coupled with chromosome congression near the spindle poles, with CENP-E serving as a crucial link between the two processes and Aurora kinases acting as the key regulatory elements.

## Results

### CENP-E is essential only for the initiation of movement during chromosome congression

To unravel the molecular and mechanical basis of chromosome congression in relation to CENP-E activity[1], we developed an assay that integrates acute protein perturbations with long-term, high-speed live-cell imaging. By implementing a live-cell imaging strategy based on LLSM we were able to simultaneously image many individual cells (1.5 mm field of view) in complete volumes (35 μm depth) with high spatiotemporal resolution (~350 nm and 2 min/image) for a prolonged duration (>24 h) and with minimal phototoxicity (Fig. 1a and Supplementary Video 1; see also Supplementary Fig. 1 for comparison with confocal microscopy). We combined this imaging strategy with an experimental design that allowed us to acutely vary the activity or presence of CENP-E in the human non-transformed cell line RPE-1. Specifically, we tested the following conditions: (1) 'CENP-E reactivated,' achieved by acute reactivation of CENP-E motor activity via washout following inhibition with the small-molecule inhibitor GSK923295[37,38]; (2) 'CENP-E depleted,' achieved by using small

interfering RNAs (siRNAs) to deplete CENP-E; (3) 'CENP-E inhibited,' achieved by continuous inhibition of CENP-E with GSK923295; and (4) 'control,' treated with just DMSO (Fig. 1b, c). We then tracked the 3D positions of centrosomes and kinetochores over time in randomly selected cells (see Methods). Immunofluorescence demonstrated efficient depletion of all target proteins analyzed throughout this study (Supplementary Fig. 2a–h).

We reasoned that if CENP-E directly drives chromosome congression, the dynamics of polar chromosome movement toward the metaphase plate would differ depending on CENP-E activity. Conversely, if CENP-E acts in processes other than force generation during congression, these dynamics should remain largely unchanged. Following reactivation of CENP-E, all polar chromosomes congressed to the metaphase plate during 60 min (Fig. 1d, Supplementary Fig. 1a, and Supplementary Video 2). Polar kinetochores approached the equatorial region in a highly uniform manner, showing a biphasic trajectory with an initial slow phase followed by a faster phase (Fig. 1d and Supplementary Fig. 1a)[19,39], accompanied by a gradual increase in interkinetochore distance (Supplementary Fig. 1b), and progressive reorientation of the sister kinetochore axis toward the main spindle axis (Supplementary Fig. 1c). These dynamics closely resembled those of polar kinetochores during early prometaphase in control cells (Fig. 1d, Supplementary Fig. 1a–c, and Supplementary Video 2). The slow phase of congression was largely absent in control cells, consistent with the near absence of persistently polar chromosomes in RPE-1 cells (Fig. 1d)[37]. We thus conclude that the CENP-E-driven congression of polar chromosomes after CENP-E reactivation recapitulates the key features of their congression during prometaphase.

How do the dynamics of chromosome congression compare between cells with the active versus perturbed CENP-E? CENP-E depletion or inhibition slowed but did not prevent congression, as most polar chromosomes aligned within 3 h (Fig. 1d; and Supplementary Video 3) consistent with prior studies using either CENP-E depletion or distinct small-molecule inhibitors[4,26,34,40–42]. Intriguingly, although movement of polar chromosomes was initiated asynchronously during prolonged mitosis (Fig. 1d), the subsequent dynamics of their movement toward the plate were strikingly similar under both active and perturbed CENP-E conditions (Fig. 1d, Supplementary Fig. 1d, and Supplementary Video 3). Congression movement was accompanied by an increase in interkinetochore distance and reorientation of the kinetochore pair toward the spindle axis in all conditions (Fig. 1d and Supplementary Fig. 1a–d). Importantly, the mean velocities of sister kinetochore movement during the 6 min prior to joining the plate, which corresponds to the major portion of the congression movement (Fig. 1d and Supplementary Fig. 1a–d), were indistinguishable across conditions irrespectively of CENP-E activity (Fig. 1e). This suggests that polar chromosomes move from the spindle poles to the equator in a manner independent of CENP-E activity in RPE-1 cells, challenging models in which CENP-E actively transports polar chromosomes to the metaphase plate.

Since we observed that congression movement remains highly similar regardless of CENP-E presence or activity, our next objective was to understand why the absence of CENP-E in RPE-1 cells leads to extensive delays in chromosome congression. We observed that when CENP-E was perturbed, only -20% of polar kinetochores initiated congression within 30 min after spindle elongation in prometaphase. By contrast, reactivation of CENP-E increased this fraction to >85%, while under control conditions all polar chromosomes initiated congression within the same 30-min period (Fig. 1f, Supplementary Fig. 1a–d, and Supplementary Videos 2 and 3). This suggests that CENP-E plays a role in the initiation of kinetochore movement. Finally, we examined whether congression dynamics depend on the duration of mitosis. We found no correlation between congression velocity and the time from mitotic entry to congression initiation across conditions

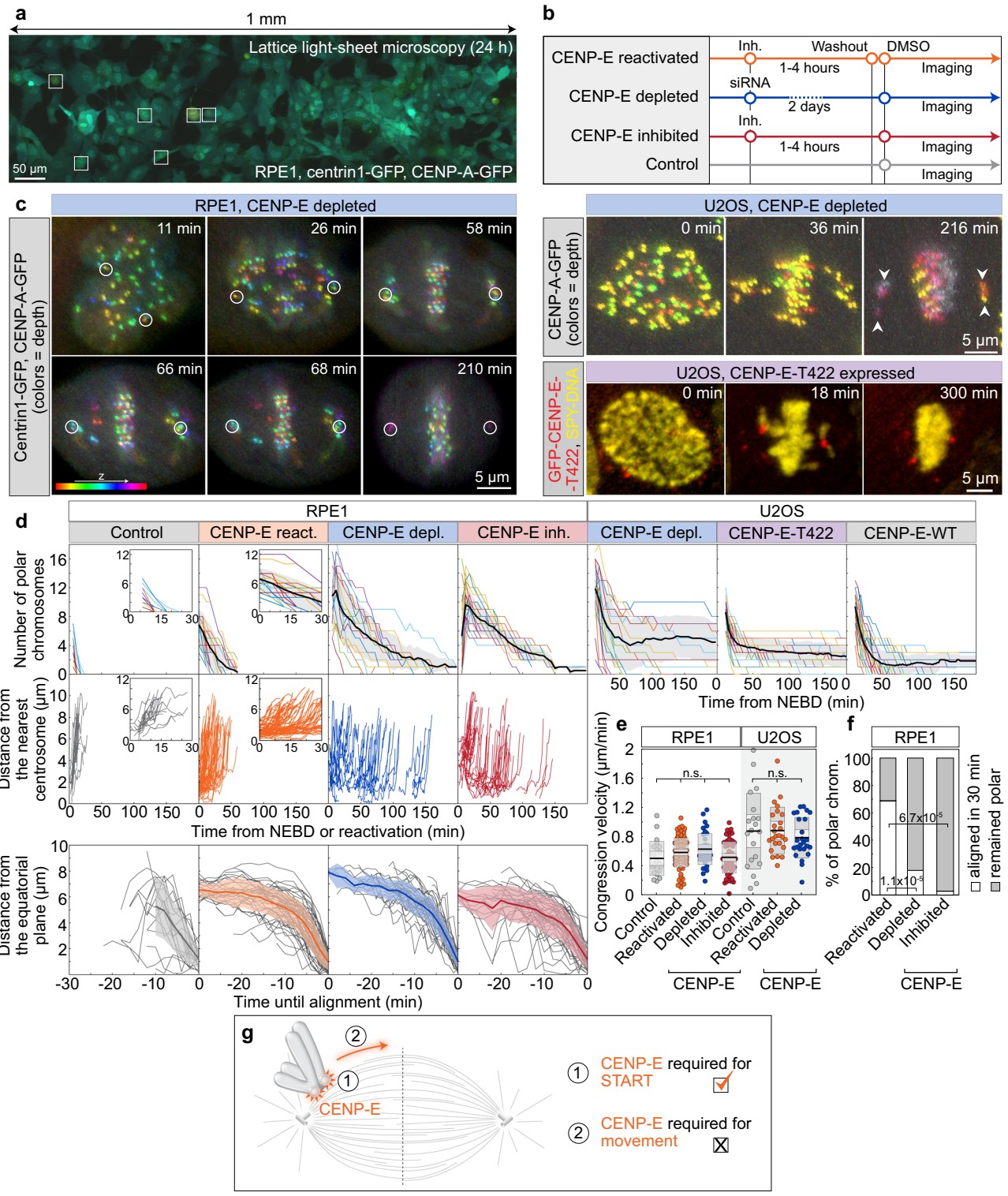

**a** 1 mm — Lattice light-sheet microscopy (24 h) — 50 µm — RPE1, centrin1-GFP, CENP-A-GFP

**b**
CENP-E reactivated — Inh. — Washout — DMSO — Imaging — 1-4 hours
CENP-E depleted — siRNA — 2 days — Imaging
CENP-E inhibited — Inh. — 1-4 hours — Imaging
Control — Imaging

**c** RPE1, CENP-E depleted — Centrin1-GFP, CENP-A-GFP (colors = depth) — 11 min, 26 min, 58 min, 66 min, 68 min, 210 min — 5 µm

U2OS, CENP-E depleted — CENP-A-GFP (colors = depth) — 0 min, 36 min, 216 min — 5 µm

U2OS, CENP-E-T422 expressed — GFP-CENP-E-T422, SPY-DNA — 0 min, 18 min, 300 min — 5 µm

**d** RPE1 — Control, CENP-E react., CENP-E depl., CENP-E inh. — U2OS — CENP-E depl., CENP-E-T422, CENP-E-WT

Number of polar chromosomes — Distance from the nearest centrosome (µm) — Time from NEBD or reactivation (min) — Time from NEBD (min) — Distance from the equatorial plane (µm) — Time until alignment (min)

**e** Congression velocity (µm/min) — RPE1: Control, Reactivated, Depleted, Inhibited — U2OS: Control, Reactivated, Depleted — CENP-E — n.s.

**f** % of polar chrom. — RPE1: Reactivated, Depleted, Inhibited — CENP-E — aligned in 30 min / remained polar — $6.7\times10^{-5}$ — $1.1\times10^{-5}$

**g** CENP-E — ① CENP-E required for START ☑ — ② CENP-E required for movement ☒

(Supplementary Fig. 1e), indicating that chromosome movement during congression is robust to the time a cell spends in mitosis.

To quantify the dynamics of chromosome congression initiation in the absence of CENP-E motor activity without directly perturbing the motor domain, we used an osteosarcoma U2OS cell line with doxycycline-inducible expression of a GFP-tagged phospho-null T422A mutant[43] (Fig. 1c), which blocks CENP-E phosphorylation by Aurora kinases[26,43]. Congression initiation timing was compared between these cells, cells expressing inducible GFP-tagged wild-type (WT) CENP-E, and CENP-E-depleted cells (Fig. 1c and Supplementary Fig. 3a).

Quantification of congression events over time revealed that T422A-expressing cells initiated congression of polar chromosomes similarly to CENP-E-depleted cells, whereas WT CENP-E expression enhanced congression initiation efficiency (Fig. 1d). These findings suggest that the T422A mutant functionally mimics CENP-E loss[43], supporting the conclusion that polar chromosome initiation can proceed independently of CENP-E activity.

To examine congression movement dynamics in transformed cells under varying levels of CENP-E, we imaged U2OS cells expressing CENP-A-GFP and measured congression velocities under untreated, CENP-E-

**Fig. 1 | CENP-E activity is required for the rapid initiation of congression but not for the subsequent movement of polar chromosomes. a** Lattice light sheet microscopy (LLSM) image of RPE-1 cells expressing centrin1-GFP and CENP-A-GFP (color coded for depth), treated with 80 nM CENP-E inhibitor (GSK-923295) for 3 h. Boxed areas highlight mitotic cells. **b** Schematic of the experimental approach to acutely modulate CENP-E activity. **c** Representative LLSM time-lapse images showing: RPE-1 cell expressing centrin1-GFP and CENP-A-GFP (color coded by depth) after CENP-E depletion (left); U2OS cell with CENP-A-GFP (color coded by depth) and polar chromosomes (arrowheads) during early anaphase (top right); and U2OS cell expressing GFP-CENP-E-T422 (red) after CENP-E depletion and SPY650-DNA staining (yellow) (bottom right). All images are maximum projections. Time 0 is mitosis onset. **d** Number of polar chromosomes in RPE-1 (left) and U2OS (right) cells (top), distance to the nearest spindle pole of polar kinetochore pairs in RPE-1 cells (middle) over time from NEBD or inhibitor reactivation and their distance from the equator until time of successful alignment (bottom). Thick lines; means; shaded areas, standard deviations. **e** Chromosome congression velocity in

the final 6 min (RPE-1) or 4 min (U2OS) before alignment for indicated treatments. Colored points represent individual cells; black lines show the mean, with light and dark gray areas marking 95% confidence intervals for the mean and standard deviation, respectively. **f** Percentage of polar kinetochore pairs that aligned (white) or remained polar (gray) within 30 min post-spindle elongation in RPE-1 under indicated conditions. **g** Schematic illustrating two hypotheses: CENP-E is required for congression initiation (1) or for continued movement (2). The supported hypothesis is indicated with a tick. Numbers: (d and e) control (15 cells, 32 kinetochores), reactivated (18, 86), depleted (18, 69), inhibited (12, 65) (RPE-1), (d) T422 (35 cells), WT (35), depleted (21) (U2OS), (e): control (30 cells, 64 kinetochore pairs), reactivated (10, 38), depleted (10, 52) (U2OS), (f) reactivated (23 cells), depleted (38), inhibited (14) (RPE-1), all from ≥3 independent biological replicates. Statistics: two-tailed ANOVA with post hoc Tukey's HSD test (**e**), pairwise two-proportion z-test, two-tailed (**f**). Symbols: n.s., $P > 0.05$; inh., inhibited; depl., depleted; chrom., chromosomes; siRNA, small interfering RNA; NEBD, nuclear envelope breakdown. Source data are provided as a Source Data file.

---

depleted, and CENP-E-inhibited conditions (Fig. 1c and Supplementary Fig. 3b, c). Consistent with our findings in RPE-1 cells (Fig. 1e and Supplementary Fig. 1b), both congression velocity (Fig. 1 and Supplementary Fig. 3d) and the extent of interkinetochore stretch (Supplementary Fig. 3e) during the final four minutes before alignment were similar across conditions with varying activity of CENP-E. Together, these results demonstrate that while CENP-E is not moving chromosomes during congression, it is essential for the timely initiation of congression in both transformed and non-transformed cell types (Fig. 1f).

### The speed of congression initiation is determined by the rate of end-on attachment formation

Since chromosome congression is preceded by an initiation phase, we hypothesized that this phase depends on end-on attachments at polar kinetochores and that CENP-E actively promotes their formation or stabilization[11] prior to chromosome movement. To test this hypothesis, we reactivated CENP-E in cells expressing the SAC protein Mad2, which accumulates on kinetochores lacking stable end-on attachments[44–47], enabling us to track changes in kinetochore microtubule attachments following acute CENP-E reactivation. Immediately after CENP-E reactivation, polar kinetochores had high levels of Mad2, indicating the absence of end-on attachments (Fig. 2a, b and Supplementary Fig. 4a, b)[4]. As a response to CENP-E reactivation, Mad2 levels on polar kinetochores gradually decreased over time (Fig. 2a, b; Supplementary Fig. 4a, b, and Supplementary Video 4), in contrast to the cells in which CENP-E was continuously inhibited and Mad2 levels remained constant (Fig. 2c). Interestingly, the drop of Mad2 on polar kinetochores started, on average, prior to their fast movement towards the equatorial plane although it continued to drop as kinetochores congressed (Fig. 2d and Supplementary Fig. 4a-c). In line with that, Mad2 dropped also on kinetochores that did not initiate movement during the imaging, though at a slower rate (Supplementary Fig. 4b). These results suggest that the formation of end-on attachments at polar kinetochores depends on CENP-E activity and occurs during the initial phase of congression, prior to the rapid movement of the kinetochore toward the metaphase plate.

If end-on conversion is coupled to congression, then congressing chromosomes would predominantly arrive at the equatorial plate already bioriented. Indeed, after CENP-E reactivation, sister kinetochores closer to the spindle pole displayed higher average Mad2 signals, which decreased as most kinetochores moved toward the metaphase plate, acquiring signal intensities comparable to those of fully aligned kinetochores (Fig. 2e, f). The Mad2 signal was negatively correlated with the distance between sister kinetochores, which serves as a readout of interkinetochore tension (Supplementary Fig. 4d). These results suggest that after CENP-E is reactivated, polar kinetochores progressively establish more end-on attachments with microtubules, generating increased tension on the kinetochores[46]. If the extent of end-on conversion is dictating dynamics of congression,

then the average congression speed over a longer period, which reflects whether or not end-on conversion occurred, is expected to be correlated with Mad2 loss. Indeed, the speed of kinetochore movement during 30 min after CENP-E reactivation was correlated with the rate of Mad2 loss (Fig. 2g), suggesting that the speed of congression initiation is dictated by the rate of formation of end-on attachments, which is reflected in the rate of Mad2 loss (Fig. 2h).

### Congressing chromosomes are characterized by the gradual stabilization of end-on attachments and ongoing fibrous corona stripping

To directly test if congression is coupled with biorientation, we performed super-resolution imaging of spindle microtubules by using STimulated Emission Depletion (STED) microscopy in CENP-E inhibited cells, and after reactivation of CENP-E at two time-points in RPE-1 and U2OS cells. This approach enabled us to quantify different types of microtubule attachments to kinetochores during chromosome congression (Fig. 3a and Supplementary Fig. 4e). The reactivation of CENP-E was accompanied by chromosome congression and a gradual transition from lateral to end-on attachments on polar chromosomes in RPE-1 (Fig. 3a, b), and U2OS cells (Supplementary Fig. 4e), while the attachment status on aligned kinetochores remained unchanged (Fig. 3c). The interkinetochore distance, along with the fraction of bioriented kinetochores, progressively increased with distance from the centrosome until reaching approximately 2.5 μm, at which point nearly all polar kinetochores were bioriented (Fig. 3d). In CENP-E-inhibited cells (Supplementary Fig. 4e), congressing chromosomes situated between the metaphase plate and the spindle pole, which represented less than 5% of all polar chromosomes, were also bioriented and stretched (Fig. 3e and Supplementary Fig. 4f). Collectively, our results demonstrate that chromosome movement during congression involves conversion from lateral to end-on kinetochore-microtubule attachments close to the spindle poles, with CENP-E playing an important role in accelerating this process.

To independently test whether stabilization and maturation of end-on microtubule attachments occur at congressing chromosomes, we stained RPE-1 cells for Astrin, a positive marker of stable end-on attachments[11,46,48,49] and CENP-E, a marker of fibrous corona[30,50]. To capture the full range of kinetochore positions from the spindle poles to the equatorial plate, we analyzed cells under CENP-E inhibition, at two time points following CENP-E reactivation, and in DMSO-treated prometaphase cells undergoing congression (Fig. 3f). We quantified the intensity of Astrin and CENP-E at all polar kinetochores and at randomly selected aligned kinetochores, together with their distance from the nearest spindle pole. Following CENP-E reactivation, Astrin levels at polar kinetochores progressively increased as kinetochores were found closer to the equatorial plane, reaching values comparable to those at aligned kinetochores (Fig. 3g). Increase in Astrin on congressing kinetochores

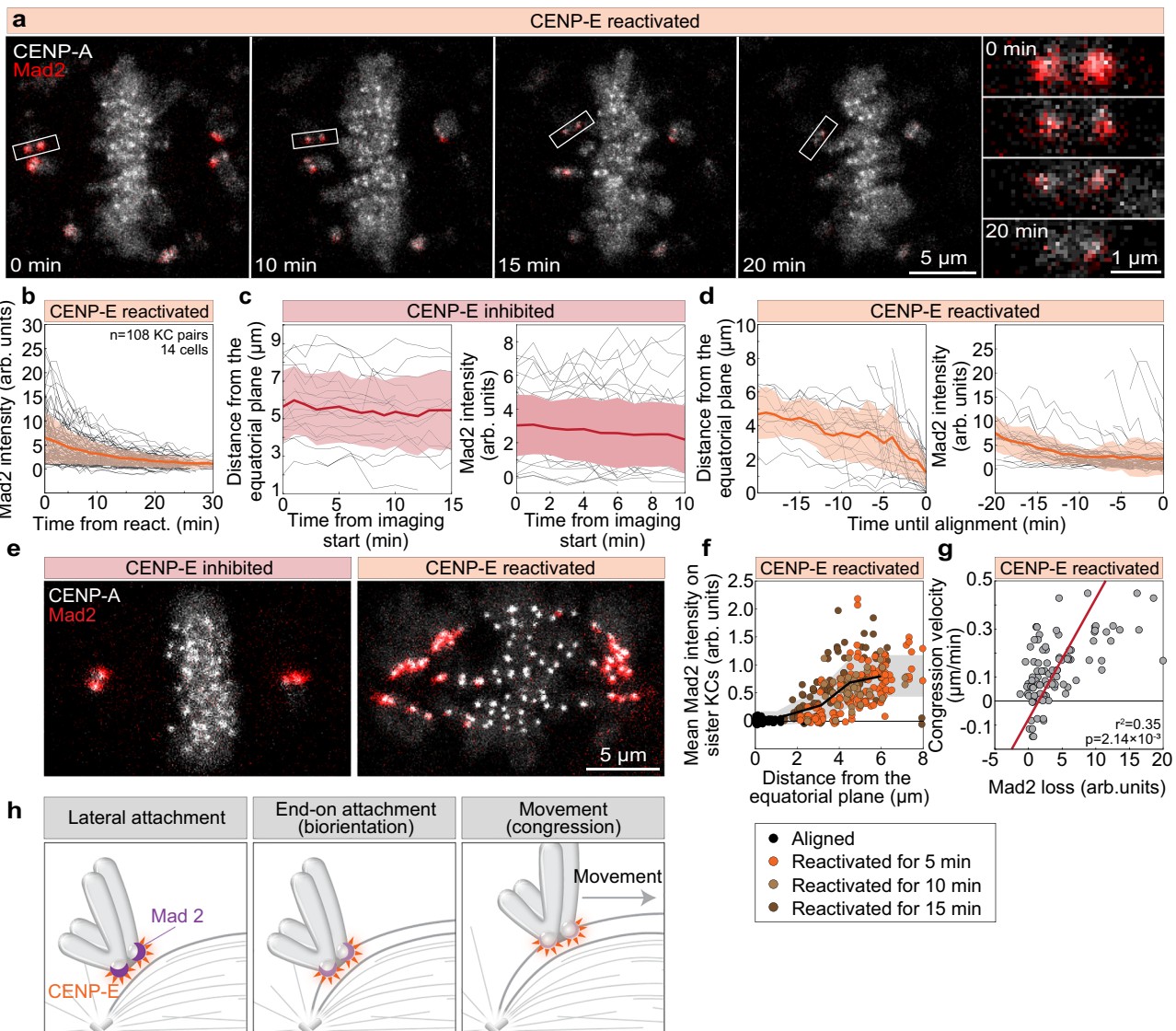

**Fig. 2 | Reactivation of CENP-E triggers Mad2 loss from polar kinetochores prior to chromosome movement. a** Representative time-lapse images of a cell expressing Mad2-mRuby (red) and CENP-A-mCerulean (gray) following CENP-E inhibitor washout; right panels show enlarged views of kinetochore pairs during congression. **b** Mad2 intensity on initially polar sister kinetochores over time post-washout of CENP-E inhibitor. **c** Distance from equator (left) and Mad2 intensity (right) of polar kinetochores over time in presence of CENP-E inhibitor. **d** Distance to equator (left) and Mad2 intensity (right) of polar kinetochores over time until their alignment. Thick lines in (**b**–**d**); means; shaded areas, standard deviations. **e** Representative images of cells fixed with or without CENP-E inhibitor showing Mad2-mRuby (red) and CENP-A-mCerulean (gray). **f** Distance to metaphase plate

versus mean Mad2 signal on kinetochores across indicated conditions; thick line: mean of binned data; gray area, standard deviation. **g** Congression velocity over 30 min after CENP-E inhibitor washout plotted against Mad2 signal loss, with regression line. Negative values reflect movement from the metaphase plate. **h** Diagram illustrating partial Mad2 loss (purple crescent) from kinetochores before rapid chromosome movement; CENP-E shown in orange. Complete Mad2 removal occurs progressively during congression. All images are maximum z-projections. Numbers: (**c**) 10 cells, 20 (left) and 56 (right) kinetochore pairs; (**d, g**) 14 cells, 54 kinetochore pairs; (**f**) 26 cells, 135 kinetochore pairs; all from ≥3 independent biological replicates. Statistics: two-tailed t-test. Symbols: arb., arbitrary; react., reactivated; KC, kinetochore. Source data are provided as a Source Data file.

mirrored both Mad2 loss (Fig. 2f) and end-on attachment maturation observed by STED microscopy (Fig. 3d) as the distance from the pole increased. Notably, by 15 min post-washout, Astrin levels at congressing kinetochores became indistinguishable from those in DMSO-treated prometaphase cells (Fig. 3g). In contrast, CENP-E levels declined as kinetochores moved toward the metaphase plate across conditions (Fig. 3h). CENP-E and Astrin display a mutually exclusive localization pattern at kinetochores: kinetochores enriched in CENP-E exhibit low Astrin levels, whereas those with high Astrin show reduced CENP-E presence (Fig. 3i). These results support a model in which chromosome congression involves the progressive capture and stabilization of end-on microtubule attachments (Fig. 3j), both following CENP-E reactivation and during unperturbed mitosis. Overall, congressing chromosomes are

biochemically characterized by intermediate levels of checkpoint proteins (e.g., Mad2), intermediate Astrin levels indicating the maturation of end-on attachments, and reduced levels of fibrous corona proteins like CENP-E, which reflects ongoing corona stripping.

## The expansion of the fibrous corona delays the initiation of chromosome congression in the absence of CENP-E

Could an inhibitory mechanism prevent the initiation of chromosome congression in the absence of CENP-E? Based on previous studies[51], we hypothesized that the fibrous corona, a dense protein meshwork that forms on kinetochores in the absence of end-on microtubule attachments[2,52], might play a role in this inhibition. To test if the corona inhibits congression, we compared the mean levels of CENP-E, used as

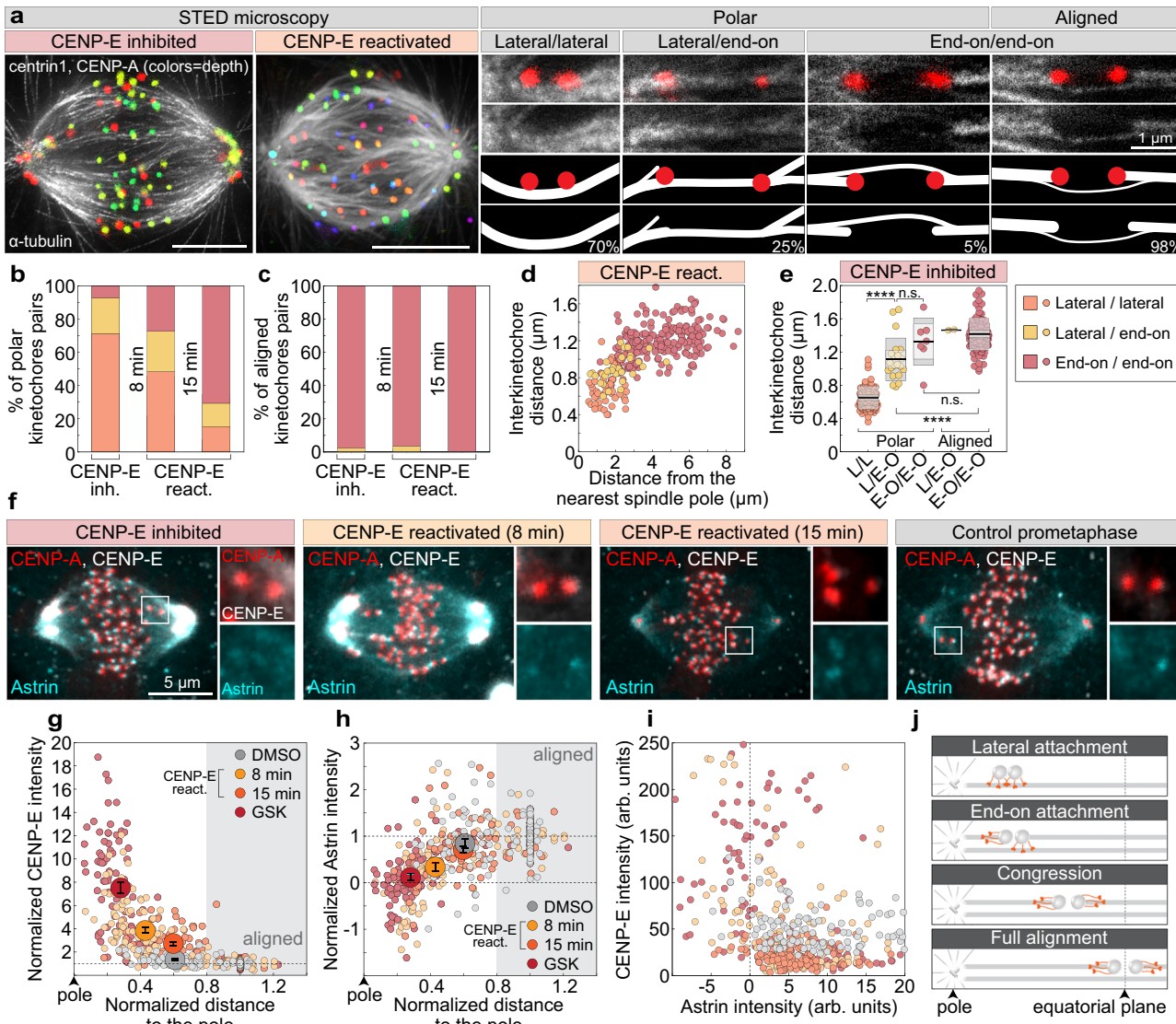

**Fig. 3 | The initiation of chromosome congression near centrosomes after CENP-E reactivation coincides with chromosome biorientation.**
**a** Representative STED images of spindles in cells expressing CENP-A-GFP and centrin1-GFP (color coded for depth, color bar in Fig. 1c), either after CENP-E inhibition or 15 min post-reactivation, immunostained for α-tubulin (gray). Maximum projections (left) and insets showing kinetochore pairs with microtubules in different attachment categories, with corresponding diagrams (right). Percentages reflect distribution of each attachment type in CENP-E-inhibited cells. Percentage of kinetochores in the indicated attachment categories for polar (**b**) and aligned (**c**) kinetochores across treatment groups (legend in **e**). **d** Interkinetochore distance versus distance from the nearest pole by attachment category (legend in **e**) following CENP-E reactivation. **e** Interkinetochore distances per attachment category in CENP-E-inhibited spindles. Colored points represent individual cells; black lines show the mean, with light and dark gray areas marking 95% confidence intervals for the mean and standard deviation, respectively. **f** Representative images of CENP-A-

GFP and centrin1-GFP (red) cells immunostained for CENP-E (gray) and Astrin (cyan) after indicated treatments, with merged projections (left) and enlarged kinetochore views (right). Astrin (**g**) and CENP-E (**h**) levels on kinetochores, normalized to CENP-A and the aligned group mean within each cell, plotted versus pole distance normalized to aligned group mean. Gray area indicates aligned region; large dots show treatment means ± SEM for each group. **i** Astrin and CENP-E levels, normalized to CENP-A, plotted against each other across the indicated treatments. **j** Schematic summarizing that lateral-to-end-on attachment transition precedes major part of congression and occurs near spindle poles following CENP-E reactivation or in its absence. All images are maximum projections. Numbers: (**b**–**d**) 69 cells, 380 kinetochore pairs; (**e**) 20 cells, 185 kinetochore pairs; (**g**–**i**) 80 cells, 505 kinetochores, all from ≥3 independent biological replicates. Statistics: two-tailed ANOVA with post hoc Tukey's HSD test. Symbols: n.s., $P > 0.05$; ****, $P \leq 0.0001$; arb,. arbitrary; inh., inhibited; react., reactivated. Source data are provided as a Source Data file.

a corona marker, and Mad2 on kinetochores following CENP-E inhibition, CENP-E depletion (note that here ZW10 was used as a corona marker, see below), and in control cells (Fig. 4a). Our analysis revealed a strong correlation between Mad2 and CENP-E levels on polar kinetochores in both control and CENP-E-inhibited cells, which did not increase with saturating concentrations of CENP-E inhibitor (Fig. 4b, c). The average levels of both proteins on polar kinetochores were significantly higher in CENP-E-inhibited cells compared to prometaphase cells (Fig. 4b, c), indicating excessive corona expansion on polar

kinetochores after CENP-E inhibition. Surprisingly, despite a significant loss of kinetochore CENP-E, Mad2 levels on polar kinetochores in CENP-E-depleted cells were comparable to those in CENP-E-inhibited cells (Fig. 4b, c). Consistent with this, ZW10, a component of the RZZ complex and corona marker[30,53], was excessively recruited to polar kinetochores relative to control prometaphase cells, especially after CENP-E depletion (Fig. 4d–f). In contrast, the level of the outer kinetochore marker Mis12 on polar kinetochores remained unchanged on polar kinetochores following CENP-E perturbations (Fig. 4d–f). These

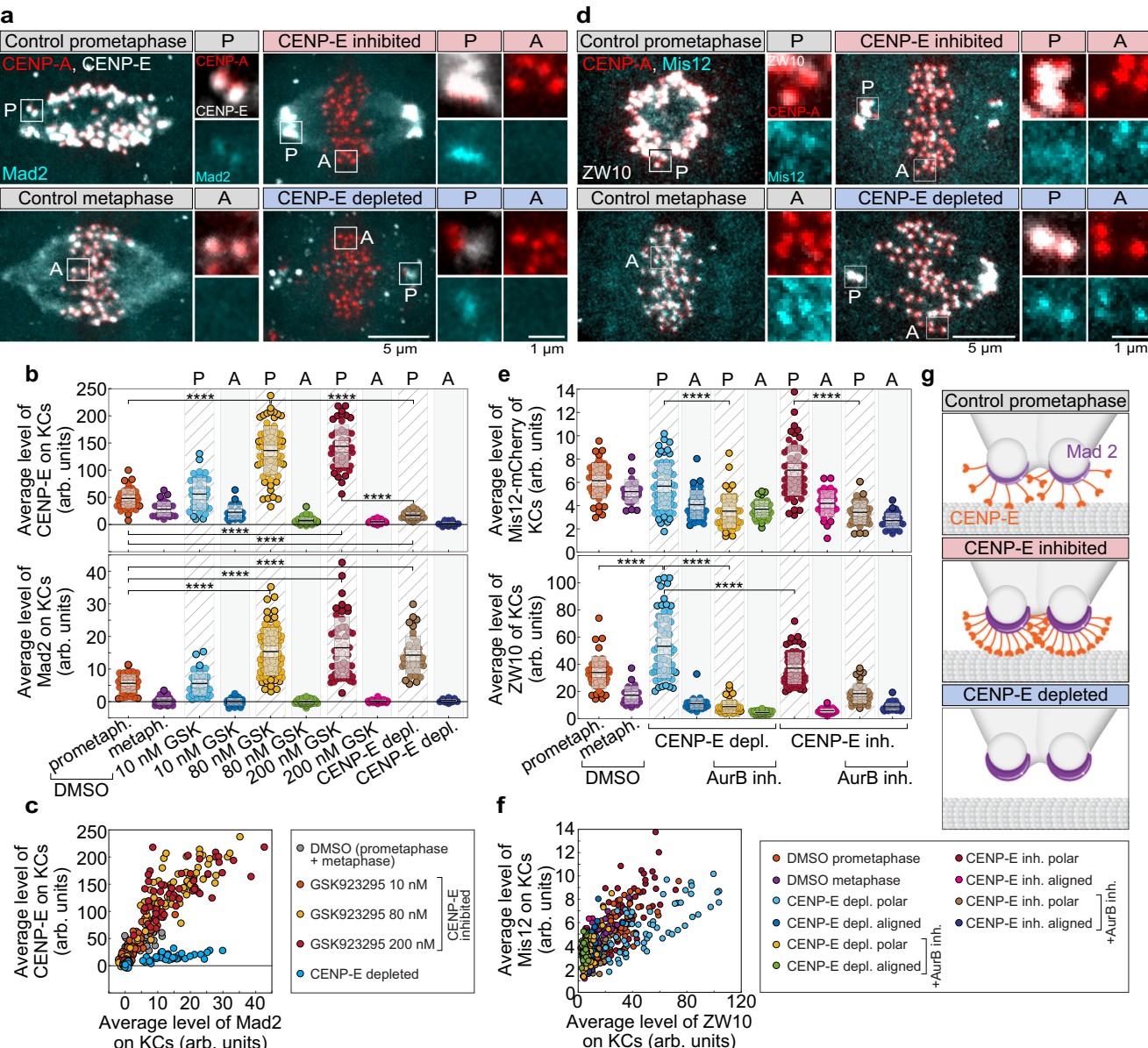

**Fig. 4 | Fibrous corona expansion on kinetochores in the absence of CENP-E or its activity inhibits the initiation of chromosome congression. a** Representative images of cells expressing Mad2-mRuby (cyan) and CENP-A-mCerulean (red), immunostained for CENP-E (gray) after indicated treatments. Merged images (left) and enlarged views of boxed kinetochore pairs (right) are shown. A, aligned; P, polar. **b** Mean CENP-E (top) and Mad2-mRuby (bottom) levels on kinetochores under the indicated treatments. GSK: GSK923295. Colored points represent individual cells; black lines show the mean, with light and dark gray areas marking 95% confidence intervals for the mean and standard deviation, respectively. **c** Correlation between average Mad2 and CENP-E levels on kinetochores across treatments. **d** Representative cells expressing Mis12-mCherry (cyan) and CENP-A-

GFP (red), immunostained for ZW10 (gray), with merged and enlarged kinetochore images. **e** Mean levels of Mis12-mCherry (top) and ZW10 (bottom) on kinetochores for each treatment. Dispersion measures as in (**b**). **f** Correlation between average ZW10 and Mis12 levels across indicated treatments. **g** Schematic of polar kinetochore states showing CENP-E (orange) laterally bound to microtubules (gray) and Mad2 (purple crescent) under indicated conditions. All images are maximum projections. Numbers: (**b**, **c**) 53 cells, 744 kinetochores; (**e**, **f**) 78 cells, 810 kinetochores, all from ≥3 independent biological replicates. Statistics: two-tailed ANOVA with post hoc Tukey's HSD test. Symbols: ****, $P \leq 0.0001$; arb., arbitrary; inh., inhibited; depl., depleted; metaph., metaphase; prometaph., prometaphase; KC, kinetochore. Source data are provided as a Source Data file.

findings suggest that without CENP-E or its activity, corona expands on polar kinetochores (Fig. 4g), contributing to the obstruction of end-on conversion[51] and chromosome congression.

## Aurora kinases delay the initiation of congression in the absence of CENP-E

After establishing that CENP-E facilitates end-on attachment formation to initiate congression, we systematically examined the molecular pathways that restrict congression when CENP-E is acutely inhibited by targeting various proteins (Fig. 5a). We hypothesized that, in addition

to the physical barrier of the expanded fibrous corona, a molecular signaling pathway inhibits the stabilization of end-on attachments in the absence of CENP-E, thereby preventing congression. Consequently, if end-on conversion is initiated in the absence of CENP-E activity, we would expect polar chromosomes to initiate congression. To test this, we inhibited Aurora B kinase that counteracts formation of end-on attachments[36], and maintenance of the Mad1:Mad2 as well as RZZ complex and fibrous corona on kinetochores[34,36,51,54]. Fascinatingly, following acute inhibition of Aurora B with 3 µM ZM-447439[55] in both CENP-E-inhibited and depleted conditions, polar chromosomes

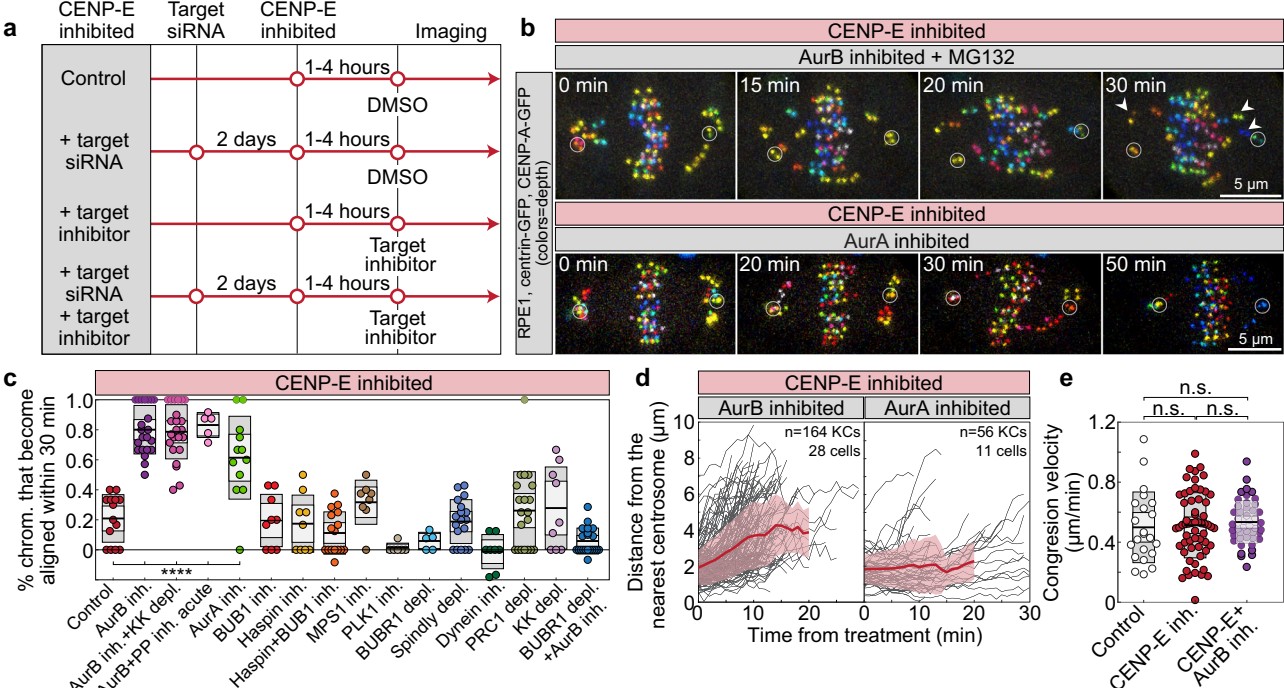

**Fig. 5 | Aurora kinases inhibit initiation of chromosome congression in the absence of CENP-E. a** Schematic of the experimental protocol used to test molecular factors that restrict chromosome congression under CENP-E inhibition. **b** Representative images of cells expressing CENP-A-GFP and centrin1-GFP (white circles mark centrioles) following acute inhibition of target proteins, as indicated, during ongoing CENP-E inhibition. Time indicates duration post-treatment. Images are color coded for depth (color bar in Fig. 1c). **c** Percentage of polar chromosomes per cell that aligned within 30 min of treatment with a CENP-E inhibitor. Colored points represent individual cells; black lines show the mean, with light and dark gray areas marking 95% confidence intervals for the mean and standard deviation, respectively. Numbers: 14, 22, 23, 5, 12, 9, 9, 17, 8, 6, 5, 18, 8, 20, 9, 21 cells, all from ≥3 independent biological replicates. Negative values indicate increased number of polar chromosomes after 30 min. **d** Distance of initially polar kinetochore pairs to the nearest centrosome over time after indicated treatment. Thick lines; means; shaded areas, standard deviations. **e** Chromosome congression velocity during the 6-min interval preceding full alignment for control, CENP-E-inhibited, and CENP-E/Aurora B co-inhibited cells. Dispersion measures as in (**c**). Numbers for (**e**) are given in legend of Fig. 1e except for third row which contains 48 kinetochore pairs from 28 cells, from ≥3 independent biological replicates. Statistics: two-tailed ANOVA with post-hoc Tukey's HSD test. Symbols: ****, $P \le 0.0001$; Inh., inhibited; depl., depleted; siRNA, small interfering RNA; PP, protein phosphatases; KK, Kid and Kif4a; KC, kinetochore; NT, non-targeting; chrom., chromosome. Source data are provided as a Source Data file.

immediately began moving, with most completing congression within 30 min after inhibitor addition (Fig. 5b–d, Supplementary Fig. 5a–f, and Supplementary Video 5). A small fraction of polar chromosomes did not reach the metaphase plate but ended their journey close to the plate (Fig. 5b, arrows, and Supplementary Fig. 5f), probably due to deficient correction of attachment errors[56]. These results imply that polar chromosomes initiate congression rapidly without CENP-E if end-on conversion is forced by the acute inhibition of Aurora B.

Chromosome congression after acute inhibition of Aurora B in the absence of CENP-E activity was not dependent on the premature onset of anaphase, which was blocked by the proteasome inhibitor MG132[57] (Supplementary Fig. 5g), nor on a lower concentration of ZM-447439 (Supplementary Fig. 5h) or the use of the highly specific Aurora B inhibitor Barasertib[58] (Supplementary Fig. 5i). Furthermore, the depletion of the cohesion release factor WAPL[59] (Supplementary Fig. 5j), the co-depletion of the chromokinesins Kid/kinesin-10 and Kif4a/kinesin-4[4] (Fig. 5c and Supplementary Fig. 5k), nor the acute inhibition of Protein phosphatases 1 and 2 A (PP1/PP2A) by Okadaic acid[60] (Fig. 5c and Supplementary Fig. 5k, l) did not stop chromosomes from initiating and completing congression after acute inhibition of Aurora B. This implies, respectively, that the loss of centromeric cohesion[61], the polar ejection forces generated by chromokinesins on chromosome arms[62,63] and the activity of phosphatases[64] are not essential for congression movement after acute inhibition of Aurora B in the absence of CENP-E.

Interestingly, chromosome congression in the absence of both CENP-E and Aurora B activity was marked by rapid movement of polar

kinetochores toward the equatorial plane, with velocity indistinguishable from other CENP-E perturbations (Fig. 5d, e and Supplementary Fig. 5a, b, m). However, a distinctive feature of congression without Aurora B activity was the absence of the initial phase of congression, similar to control cells or, to a lesser degree, CENP-E-reactivated cells (compare Supplementary Fig. 5a, b with Fig. 1d). This suggests that only the initiation of congression is inhibited by Aurora B activity, indicating that CENP-E initiates chromosome congression by downregulating Aurora B activity.

To gain further insight into the signaling mechanisms that limit chromosome congression in the absence of CENP-E, we co-inhibited or depleted various molecular regulators implicated in the establishment of biorientation or in Aurora B recruitment[65] under CENP-E inhibited condition. Interestingly, acute inhibition of Aurora A kinase, which shares specificity for kinetochore targets with Aurora B[66], induced congression of polar chromosomes in the absence of CENP-E, although this occurred more slowly than with acute Aurora B inhibition (Fig. 5b–d and Supplementary Video 5). On the other hand, acute inhibition and co-inhibition of Budding uninhibited by benzimidazoles 1 (Bub1) or Haspin, factors essential for the recruitment of Aurora B to the outer and inner centromere, respectively[67–72], did not induce chromosome congression in the absence of CENP-E activity, as well as inhibition of Polo-like kinase 1 (Plk1), and depletion of Bub1-related (BubR1) pseudokinase (Fig. 5c and Supplementary Fig. 5n). The only other kinase whose inhibition produced a minor yet statistically nonsignificant increase in congression events relative to the control group was Monopolar spindle 1 (Mps1) (Fig. 5c and Supplementary

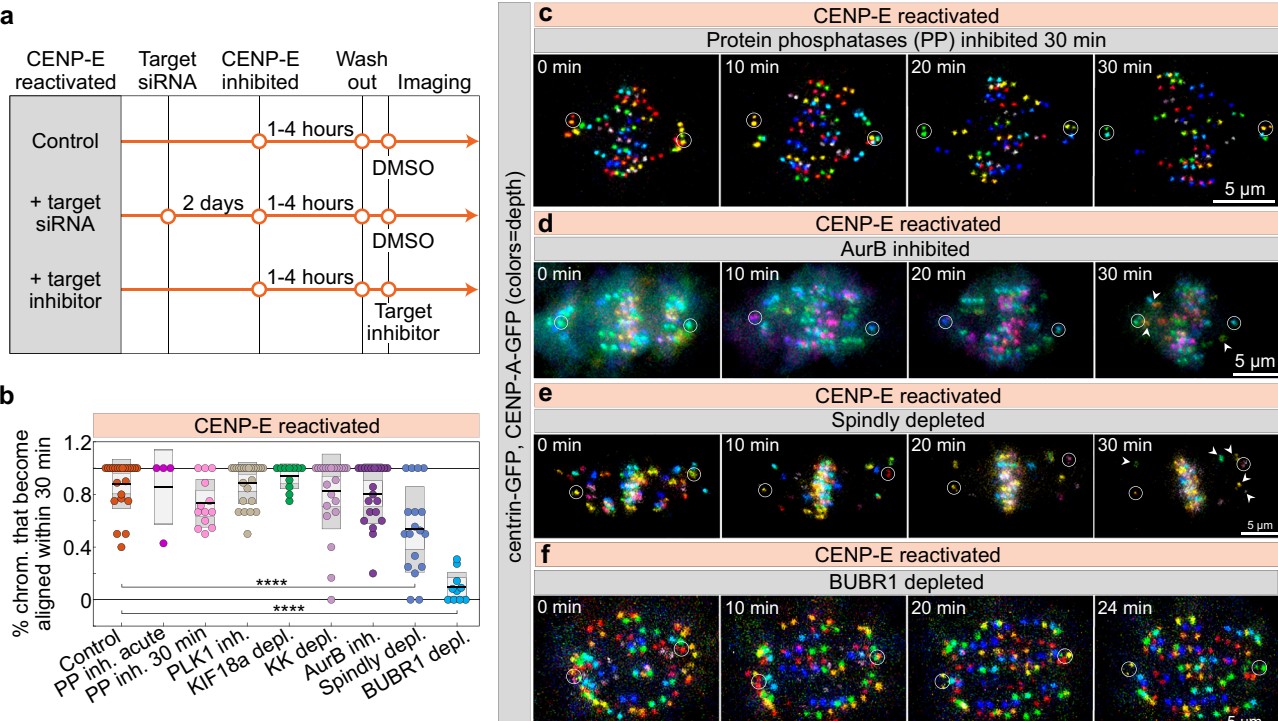

**Fig. 6 | Lateral attachments and BubR1 are required for CENP-E-mediated congression initiation. a** Experimental scheme to test factors that regulate polar chromosome congression following washout of a CENP-E inhibitor. **b** Percentage of polar chromosomes per cell that successfully aligned within 30 min of washout under the indicated treatments. Colored points represent individual cells; black lines show the mean, with light and dark gray areas marking 95% confidence intervals for the mean and standard deviation, respectively. Numbers: 23, 4, 13, 24, 11, 23, 21, 18, 10 cells, all from ≥3 independent biological replicates. Representative images of CENP-A-GFP and centrin1-GFP expressing RPE-1 cells (centrioles circled)

over time after washout of CENP-E inhibitor GSK-923295 in phosphatase-pre-inhibited (Okadaic acid) (**c**), Aurora B acutely inhibited (ZM447439) (**d**), Spindly-depleted (**e**) and BuBR1-depleted (**f**) cells. Images are maximum projections color-coded by depth (color bar in Fig. 1c). Arrowheads mark polar kinetochores failing to congress. Statistics: two-tailed ANOVA with Tukey's HSD post hoc test. Symbols: ****, $P \leq 0.0001$; inh., inhibited; depl., depleted; siRNA, small interfering RNA; PP, protein phosphatases; KK, Kid and Kif4a; chrom., chromosome. Source data are provided as a Source Data file.

Fig. 5n). Altogether, these results suggest that centrosome-localized Aurora A and the kinetochore-localized Aurora B are limiting the initiation of congression in the absence of CENP-E.

The delay in the initiation of chromosome congression in the absence of CENP-E could be due to the dominant activity of other motor proteins in the spindle. However, perturbations of microtubule motor proteins, including minus-end directed dynein[73–75], as well as plus-end directed Kid, Kif4a, Eg5, and the crosslinker of antiparallel microtubules PRC1[76,77], revealed that neither their indirect nor direct perturbations induced congression of polar chromosomes in the absence of CENP-E (Fig. 5c and Supplementary Fig. 5n, o). These results suggest, respectively, that the poleward pulling forces exerted by kinetochore dynein, the activity of chromokinesins, the distance between the plus ends of microtubules nucleated in one spindle half and the opposite spindle pole, and the thickness of antiparallel microtubule bundles to which kinetochores are initially laterally attached[78], are not major obstacles to congression in the absence of CENP-E activity. Instead, the primary barrier to congression without CENP-E is the activity of Aurora kinases.

To determine if BubR1, a known interactor of PP2A and CENP-E[36,79–81], is required for the initiation of chromosome congression after Aurora B inhibition, we acutely inhibited Aurora B in cells lacking CENP-E activity and BubR1. Strikingly, polar chromosome congression was completely blocked in these cells, unlike in cells with intact BubR1 (Fig. 5c). These results indicate that BubR1 is crucial for the initiation of chromosome congression, independent of its role in CENP-E recruitment[81], and that its absence cannot be compensated for by the inhibition of Aurora kinases.

Finally, we directly tested the role of corona expansion in inhibiting congression initiation in the absence of CENP-E by quantifying ZW10 levels on polar kinetochores following CENP-E perturbations and acute Aurora B kinase inhibition. We found that acute Aurora B inhibition reduced ZW10 levels on polar kinetochores to the levels observed on aligned kinetochores, both after CENP-E inhibition and depletion (Fig. 4e, f). These findings support our conclusion that, in the absence of CENP-E or its activity, Aurora B-dependent fibrous corona expansion inhibits the initiation of chromosome congression.

## CENP-E initiates chromosome congression in a BubR1-dependent manner

We next used the acute reactivation of CENP-E, combined with the acute inhibition of other target molecules, as a tool to identify which molecular factors regulate chromosome congression both upstream and downstream of CENP-E (Fig. 6a). As a first hypothesis, we assumed that the role of CENP-E in the initiation of chromosome congression reflects the recruitment of protein phosphatases to the outer kinetochore, either directly through interaction with PP1 or through its interaction with BubR1-PP2A[26,82]. However, acute or 30-min pre-inhibition of PP1/PP2A activities by Okadaic acid (see "Methods") did not prevent chromosome congression after reactivation of CENP-E (Fig. 6b, c, Supplementary Fig. 6a, b, and Supplementary Video 5), similar to earlier findings showing that acute PP1 perturbation in HeLa cells during prometaphase did not prevent polar chromosome congression[26]. Congression after CENP-E reactivation was also successful without the activities of Plk1 kinase, which regulates microtubule dynamics and recruits PP2A to BuBR1[80,83], Kif18a/kinesin-8,

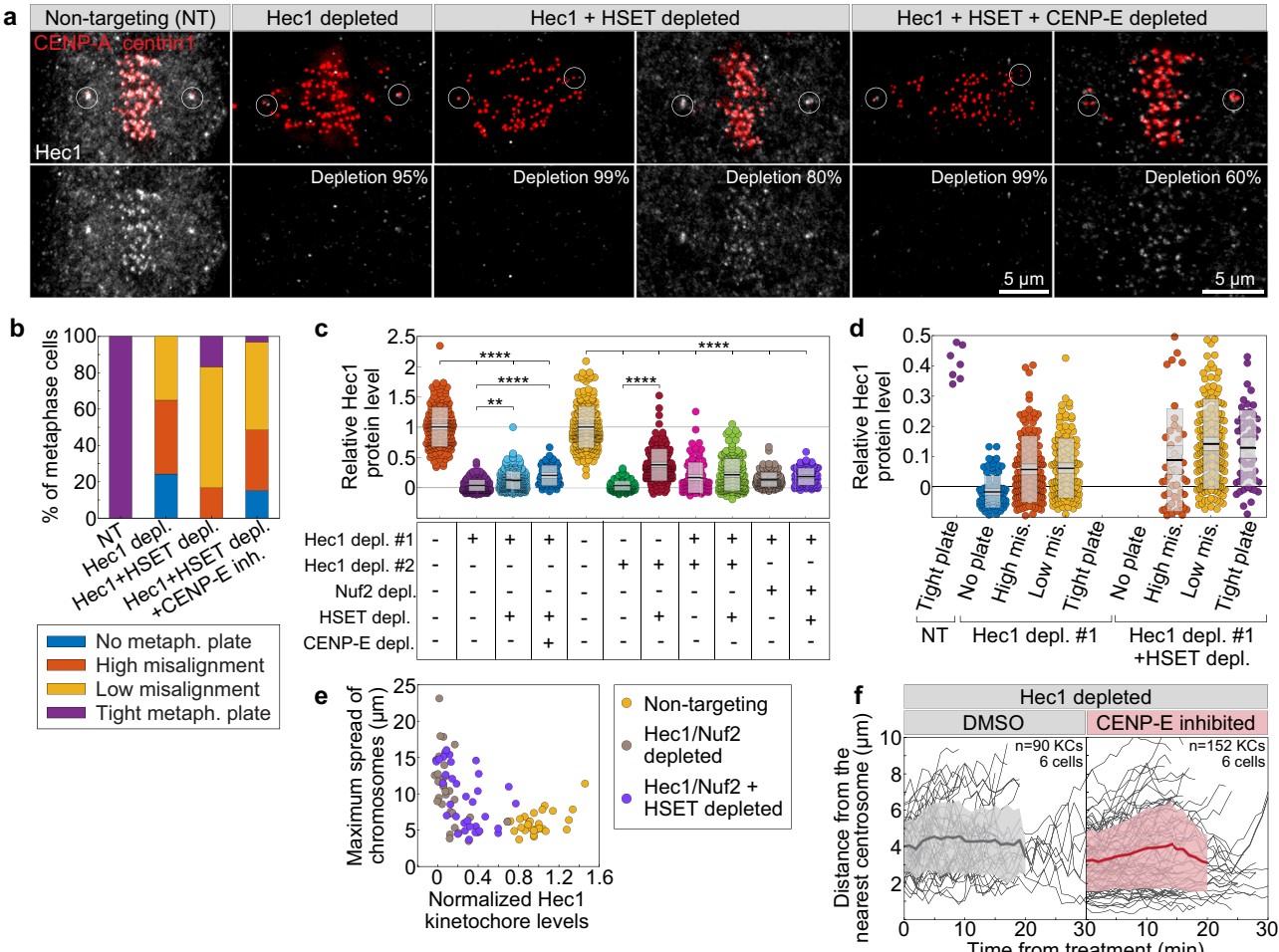

**Fig. 7 | CENP-E has limited ability to initiate chromosome movement when end-on attachment formation is compromised. a** Representative images of cells expressing CENP-A-GFP and centrin1-GFP (red) immunostained for Hec1 (gray) with circled centrioles, under indicated treatments. **b** Percentage of metaphase cells showing varying degrees of chromosome misalignment (legend) per treatments. Average Hec1 levels on kinetochores normalized to non-targeting controls across treatments (**c**) and combined with alignment categories (**d**, legend from **b**). Colored points represent individual cells; black lines show the mean, with light and dark gray areas marking 95% confidence intervals for the mean and standard deviation, respectively. **e** Hec1 levels versus maximum chromosome spread per cell across treatments (legend). **f** Distance over time from initially polar kinetochores to nearest centrosome under indicated treatments. Thick lines; means; shaded areas, standard deviations. Numbers: (**b**) 31, 27, 24, 33 cells; (**c**) 1668 kinetochores from 153 cells; (**d**) 733 kinetochores from 64 cells; (**e**) 97 cells, all from ≥3 biological replicates. All images are maximum projections. Statistics: two-tailed ANOVA with Tukey's HSD post hoc test. Symbols: **, $P \leq 0.01$; ****, $P \leq 0.0001$; inh., inhibited; depl., depleted; metaph., metaphase; NT, non-targeting; mis., misaligned. Source data are provided as a Source Data file.

which promotes chromosome alignment at the metaphase plate[84], or the chromokinesins Kid and Kif4a and thus the polar ejection forces they generate[62] (Fig. 6b and Supplementary Fig. 6c).

Phosphorylation of CENP-E by Aurora kinases has previously been reported as crucial for CENP-E's motor activity and for the congression of polar chromosomes[26]. However, the majority of polar chromosomes congressed following CENP-E reactivation even when Aurora B kinase was acutely inhibited (Fig. 6b, d). Similar to the effect of acute Aurora B inhibition when CENP-E is inactive (Fig. 5b, arrows), chromosome congression after Aurora B inhibition in the presence of active CENP-E was occasionally accompanied by incomplete congression of a small number of polar chromosomes (Fig. 6d, arrows). This suggests that incomplete congression of polar chromosomes may result from error correction defects, such as unrepaired merotelic attachments, a known consequence of Aurora B inhibition[56]. In summary, the majority of polar chromosomes can initiate congression when Aurora B is fully inhibited, regardless of CENP-E activity.

It remains unclear whether CENP-E activity requires the presence of lateral attachments to mediate congression, as would be expected if CENP-E functions through the establishment or stabilization of end-on attachments. To increase the number of polar chromosomes with prematurely stabilized end-on attachments, we reactivated CENP-E in cells depleted of Spindly, a dynein adaptor at the kinetochore[4,74]. The proportion of polar chromosomes that congressed after CENP-E reactivation was significantly lower in Spindly-depleted cells compared to controls (Fig. 6b, e). This suggests that CENP-E cannot initiate the congression of polar kinetochores with stable end-on attachments, or that Spindly contributes to recruitment of CENP-E to kinetochores, as recently reported[30].

Finally, we tested the requirement for BubR1 in CENP-E-mediated congression of polar chromosomes by depleting BubR1, inhibiting CENP-E, and treating cells with MG132 to prevent the premature onset of anaphase that is observed after BuBR1 depletion[85]. Reactivation of CENP-E in BubR1-depleted cells did not trigger congression of polar chromosomes, unlike in cells with intact BubR1 (Fig. 6b, f, Supplementary Fig. 6b and Supplementary Video 5). This finding, along with our observation that BubR1 is required to counteract Aurora B independently of its role in CENP-E recruitment (Fig. 5c), suggests that BubR1 is essential for the initiation of chromosome congression.

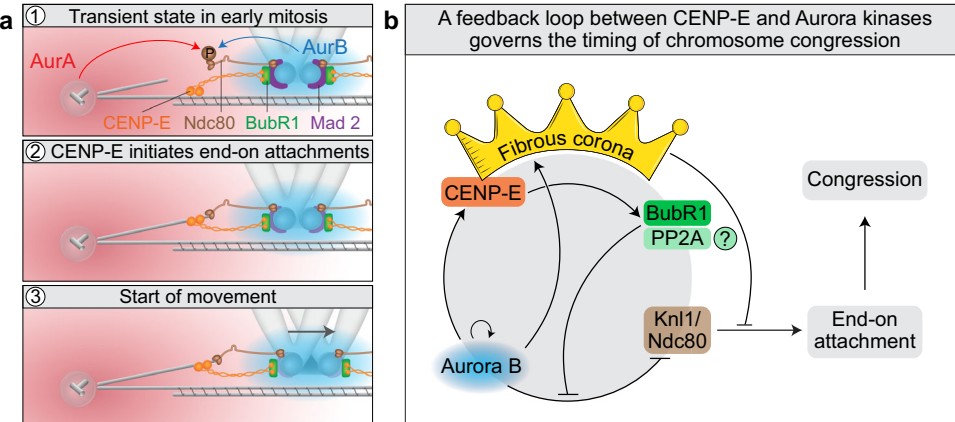

**Fig. 8 | Model of chromosome congression regulation mediated by a negative feedback loop between Aurora B kinase and CENP-E/BubR1. a** Aurora kinases inhibit congression initiation by phosphorylating the Ndc80 tail near centrosomes (1). On polar kinetochores, CENP-E–BubR1 facilitates early end-on attachment formation near the Aurora A gradient (2), triggering a decline in Aurora B activity, loss of Mad2 from the kinetochore, and stabilization of Ndc80-microtubule binding, all preceding fast kinetochore movement (3). **b** This establishes a negative feedback loop involving Aurora B, CENP-E, BubR1-PP2A, the fibrous corona, and outer kinetochore proteins (Knl1 and Hec1/Ndc80). The loop self-limits by promoting end-on attachment and initiating chromosome congression, which in turn reduces the upstream signals that sustain it. Lines with arrows indicate activation, and blunt lines indicate deactivation by phosphorylation or direct physical inhibition. Initially, polar chromosomes form lateral kinetochore-microtubule attachments that fail to convert to stable end-on attachments without CENP-E due to high Aurora B activity. Aurora B phosphorylates Knl1 and Hec1 to prevent end-on conversion and maintains the expanded fibrous corona (depicted as a crown), which further inhibits attachment stabilization. Aurora B also activates CENP-E via phosphorylation. We propose (indicated by a question mark) that activated CENP-E interacts with BubR1 and possibly PP2A to counteract Aurora B-mediated phosphorylation, enabling stabilization of initial end-on attachments, progressive corona disassembly, and chromosome congression toward the spindle midplane.

## CENP-E initiates chromosome congression in a Hec1-dependent manner

To challenge our model that congression requires biorientation, we investigated whether CENP-E can facilitate chromosome movement independently of end-on attachments in cells depleted of Hec1 and KifC1/HSET, a condition reported to permit chromosome pseudoalignment in the absence of end-on attachments[23]. We then combined this approach with CENP-E perturbations and analyzed the extent of chromosome misalignment and Hec1 kinetochore levels in individual mitotic cells using immunofluorescence. Our results show that chromosome alignment improved after HSET depletion in Hec1-depleted cells, while additional CENP-E depletion led to severe misalignment (Fig. 7a, b), consistent with previous findings[23]. Notably, the effectiveness of chromosome alignment strongly correlated with residual Hec1 levels at kinetochores, which were generally higher in Hec1- and HSET-codepleted cells than in cells depleted of Hec1 alone (Fig. 7c–e and Supplementary Fig. 6d). Importantly, cells with minimal Hec1 at the kinetochores exhibited significant chromosome misalignment, regardless of additional HSET or CENP-E co-depletions (Fig. 7a, e). We note that extensive misalignment in cells with minimal Hec1 could be exacerbated by extensive mitotic prolongation. In live-imaged Hec1-depleted cells, the movement of polar chromosomes was unaffected by additional inactivation of CENP-E (Fig. 7f), consistent with previous observations in HeLa cells[35]. These findings suggest that a low level of Hec1, about 10–15% of the levels in control cells, is sufficient to promote congression, and that CENP-E's role in chromosome congression is promoted by the presence of Hec1 on kinetochores.

## Discussion

For nearly two decades, the prevailing model has held that plus-end-directed kinetochore motors, such as CENP-E, transport chromosomes to the spindle equator independently of biorientation[19], inspiring numerous models of chromosome movement and its regulation[4,21,23,26,27,43]. Here, we provide systematic evidence supporting an alternative model where the bulk of chromosome movement during congression depends on biorientation, consistent with earlier proposals[86,87]. Our conclusions are based on key findings: (1)

once chromosome movement is initiated, it is unaffected by the presence or activity of CENP-E in both transformed and non-transformed cells; (2) end-on attachments form and the Mad2 signal decreases on polar kinetochores before their rapid movement toward the equator; (3) CENP-E levels decline early on congressing kinetochores, reflecting corona stripping that requires end-on microtubule attachments[2]; (4) all congressing chromosomes are bioriented a few microns from the centrosome, as shown by STED microscopy and Astrin labeling; (5) CENP-E has limited ability to drive chromosome movement without end-on attachments or if these attachments stabilize prematurely near the pole; (6) congression can initiate without CENP-E when Aurora kinases are inhibited, which promotes biorientation; and (7) BubR1, a known promoter of end-on conversion[88], is essential for congression independently of its role in recruiting CENP-E.

Mechanistically, we propose that CENP-E stabilizes nascent end-on kinetochore-microtubule attachments, especially near spindle poles where high Aurora A activity likely amplifies Aurora B signaling, promoting microtubule detachment in the absence of CENP-E[89] (Fig. 8a). This model is supported by in vitro evidence showing that Ndc80 and CENP-E together are sufficient for lateral-to-end-on transitions[90]. While CENP-E has been proposed to maintain lateral attachments in cells[11], our findings suggest it may also directly promote end-on conversion and biorientation, independently of its gliding activity. Although non-gliding roles for CENP-E have been proposed[21,91,92], their contribution to stabilizing end-on attachments during congression remains to be tested. For example, CENP-E-mediated transport of laterally attached microtubules along the kinetochore surface that delivers their plus-ends near kinetochores[33,92], could facilitate KMN network function and stabilize end-on attachments. Recent studies reveal that CENP-E exists in distinct pools at the kinetochore and fibrous corona, recruited by BubR1 and the RZZ complex, respectively[30,32,81]. Given that congression initiation fully depends on BubR1, we propose that kinetochore-localized CENP-E primarily drives this process.

We propose that the main barrier to congression initiation in the absence of CENP-E is Aurora B kinase activity (Fig. 8b). Aurora B limits the stability of end-on attachments by phosphorylating KMN

network[36,93] and regulating the RZZ complex and fibrous corona expansion[51,54]. Aurora A also directly phosphorylates KMN network near centrioles[66]. By preventing erroneous end-on attachments near centrosomes[94], Aurora kinases and the back-to-back arrangement of kinetochores[4,66], ensure that congression initiation is not blocked. Aurora kinases also phosphorylate and activate CENP-E near centrioles[26,43] (Fig. 8b). We speculated that activated CENP-E interacts with BubR1-PP2A[88], and that PP2A downregulates Aurora B kinase, leading to dephosphorylation of its substrates at the outer kinetochore. This would stabilize early end-on attachments on polar kinetochores, trigger the onset of Mad2 loss and corona stripping, and thereby initiate congression (Fig. 8b). However, direct acute phosphatase as well as Plk1 inhibition, which should block phosphatase activity via BubR1[80], did not prevent efficient congression, suggesting that phosphatases are not essential. Future studies using separation-of-function mutants of CENP-E and BubR1 could clarify this mechanism.

Aurora kinases not only phosphorylate components of the KMN network but also inhibit end-on attachment stabilization by promoting expansion of the fibrous corona on unattached chromosomes (Fig. 8b). Prior models proposed that competition between the corona, which can nucleate, sort, or capture microtubules[33,50,53], and the Ndc80 complex regulates microtubule engagement and end-on conversion[51]. This is supported by several key observations. First, Aurora B inhibition leads to near-complete corona removal[48], which rapidly initiates chromosome congression. Second, CENP-E and Astrin exhibit mutually exclusive localization at kinetochores, indicating a switch-like mechanism tied to end-on attachment formation. Third, in cells expressing constitutively dephosphorylated Hec1, polar chromosomes only congress when CENP-E is depleted and not when it is inhibited[89], indicating that hyperexpanded coronas induced by CENP-E inhibition can block congression even when KMN microtubule affinity is enchanced. However, perturbations of Spindly or Mps1, key corona regulators[53], affect congression less than Aurora kinase inhibition, suggesting that KMN network phosphorylation represents the primary barrier to end-on attachment stabilization, unless corona is excessively enlarged, in which case it can actively inhibit attachment formation.

Our model predicts that chromosome congression can initiate without CENP-E activity when the KMN network has high microtubule-binding affinity, promoting the stabilization of nascent end-on attachments and stripping of the fibrous corona (Fig. 8b). We demonstrated this mechanism on polar chromosomes both after Aurora kinase inhibition in the absence of CENP-E and after CENP-E reactivation in the presence of Aurora kinase activity. We have also shown that a similar mechanism occurs during unperturbed mitosis, where kinetochores distant from centrosomes can stabilize end-on attachments independently of CENP-E[89]. Kinetochores unable to stabilize end-on attachments early in prometaphase are likely moved poleward by corona-associated dynein[28] and microtubule flux[43]. In intact cells with active CENP-E, the motor likely enhances this process at all kinetochores contributing to rapid biorientation and congression[33]. This is supported by findings that kinetochore fibers of aligned chromosomes are thinner when CENP-E is disrupted[86,95]. Our model also aligns with observations that constitutively dephosphorylated Hec1, which mimics high microtubule affinity, does not block congression[93,96], and that end-on attachments can form in bipolar spindles even when lateral attachments are enforced by constitutive expression of outer kinetochore-localized Aurora B[36].

We do not rule out that CENP-E may facilitate chromosome movement near the pole by gliding polar kinetochores along highly detyrosinated microtubules, helping them move away from the Aurora A activity gradient[4,27]. However, in healthy cells, chromosomes rarely approach centrosomes during mitosis[33,37] and thus avoid the region of high Aurora A activity. In contrast, in cancer cells, polar chromosomes often move close to centrosomes early in mitosis, causing substantial congression delays[8,97]. We propose that this spatial avoidance in healthy cells supports efficient chromosome biorientation and congression. As detailed in the accompanying manuscript[89], centrosomal Aurora A may provide a key spatial cue that defines the requirement for CENP-E, since CENP-E acts to counterbalance Aurora kinase activity to initiate chromosome movement.

What determines the direction of movement for polar chromosomes, causing them to migrate toward the equator rather than the spindle pole? Building on our proposed mechanism for chromosome centering during metaphase[98], and a theoretical model of prometaphase congression[99], we suggest that this direction is set by an asymmetry in the poleward flux of kinetochore microtubules. Specifically, kinetochore microtubules oriented toward the distant pole are expected to exhibit faster poleward flux than those facing the nearer pole, thereby producing a net force that directs chromosome movement toward the spindle center. Future research will clarify the relevance of this model to polar chromosome congression and identify the factors that control their directional movement.

The relationship between chromosome congression and biorientation has remained unclear. It has been well established that maintenance of chromosome position at the equator after congression requires stable kinetochore-microtubule attachments and biorientation[1,13]. In contrast, congression was thought to occur via two distinct pathways, either via biorientation or through CENP-E-mediated mechanisms[1,4,13,19]. Our results support a model in which CENP-E links congression to biorientation near spindle poles, reinforcing a unified view of chromosome movement. In conclusion, our model, together with findings from the accompanying manuscript[89], provides a molecular framework for the spatial and temporal coordination of congression and biorientation, with implications for cellular systems in which these processes are highly deregulated, such as cancer cells.

## Methods
### Cell lines and culture
All reagents and tools used in this manuscript, along with their details, are provided in Supplementary Table 1. Experiments were carried out using human hTERT-RPE-1 (hTERT-immortalized retinal pigment epithelium) cells stably expressing CENP-A-GFP, human hTERT-RPE1 cells stably expressing both CENP-A-GFP and centrin1-GFP and human hTERT-RPE1 cells stably expressing CENP-A-GFP and Mis12-mCherry, all courtesy of Alexey Khodjakov (Wadsworth Center, New York State Department of Health, NY, USA), human hTERT-RPE1 cells stably expressing CENP-A-mCerulean and Mad2-mRuby, courtesy of Jonathon Pines (Institute of Cancer Research, London, UK), human HeLa (cervical carcinoma patient) cells expressing GFP-Hec1-9A, courtesy of Geert Kops (Hubrecht Institute, Utrecht, The Netherlands), human U2OS (osteosarcoma patient) cells with inducible expression of GFP-tagged wildtype (WT) CENP-E or a phosphonull CENP-E mutated at the AurA/B-specific phospho-site Threonine 422 (T422A), and human U2OS cells expressing CENP-A-GFP, were a gift from Marin Barišić (Danish Cancer Institute, Copenhagen, Denmark). All cell lines were cultured in flasks in Dulbecco's Modified Eagle's Medium with 1 g/L D-glucose, pyruvate and L-glutamine (DMEM, Thermo Fisher, 11995065), supplemented with 10% (vol/vol) heat-inactivated Fetal Bovine Serum (FBS, Sigma-Aldrich, 16000044) and penicillin (100 IU/mL)/streptomycin (100 mg/mL) solution (Gibco, 15140122). The cells were kept at 37 °C and 5% $CO_2$ in a humidified incubator (Galaxy 170 S $CO_2$, Eppendorf) and regularly passaged at the confluence of 70–80%. None of the cell lines were authenticated. All cell lines have also been tested for mycoplasma contamination once a month by examining samples for extracellular DNA staining with SiR-DNA (100 nM, Spirochrome, SC007) or Hoechst 33342 dye (1 drop/2 ml of NucBlue Live

ReadyProbes Reagent, H3570, Thermo Fisher Scientific) and have been confirmed to be mycoplasma free.

## Sample preparation and siRNAs

At 80% confluence, the DMEM was removed from the flask and the cells were washed with 5 ml of phosphate buffered saline (PBS). Then, 1 ml 1% trypsin/ethylenediaminetetraacetic acid (EDTA, Biochrom AG, L 2153) was added to the flask and cells were incubated at 37 °C and 5% $CO_2$ in a humidified incubator for 5 min. After incubation, trypsin was blocked by adding 4 ml of DMEM. For the RNAi experiments, cells were seeded to reach 60% confluence the next day and cultured on 35 mm uncoated plates with 0.17 mm (#1.5 coverglass) glass thickness (Mat-Tek Corporation) in 1 ml of DMEM with the supplements described above. After one day of growth, cells were transfected with either targeting or non-targeting siRNA constructs which were diluted in Opti-MEM (31985062, Thermo-Fisher Scientific) to a final concentration of 100 nM in the medium with cells. All transfections were performed 48 h before imaging using Lipofectamine RNAiMAX Reagent (13778150, Invitrogen) according to the instructions provided by the manufacturer, unless otherwise indicated. Codepletions of Hec1 or Nuf2 and HSET and CENP-E in RPE-1 cells were performed for two times in two subsequent days by the same protocol. After four hours of siRNA treatment, the medium was changed to the prewarmed DMEM. Proteasome inhibitor MG-132 (Merck, M7449, IC50 value 100 nM) was used at a final concentration of 1 μM where indicated.

The siRNA constructs used were: human CENP-E ON-TARGETplus SMART pool siRNA (L-003252-00-0010, Dharmacon), human PRC1 ON-TARGETplus SMART pool siRNA (L-019491-00-0020, Dharmacon), human custom-made Ndc80/Hec1 siRNA (oligo #1, sequence: 5′-GAAUUGCAGCAGACUAUUA-3′, Dharmacon), human MISSION esiRNA kntc2 (NDC80) (oligo #2, EHU042171-20UG, Merck), human ON-TARGETplus SMART pool NUF2 siRNA (L-005289-00-000, Dharmacon), human ON-TARGETplus SMART pool KIFC1 (HSET) siRNA (L-004958-0010, Dharmacon), human KIF4A siRNA (sc-60888, Santa Cruz Biotechnology), human Kif18A siRNA (Ambion, #Cat 4390825, ID:s37882), human Kif22/Kid siRNA (Ambion, #Cat 4392420, ID:s7911), human WAPL ON-TARGETplus SMART pool siRNA (J-026287-10-0010, Dharmacon), human BUB1B/BuBR1 ON-TARGETplus SMART pool siRNA (L-004101-00-0005, Dharmacon), human Spindly siRNA (Sequence antisense: 5′- GAAAGGGUCUCAAACUGAA-3′, dTdT overhangs, Sigma-Aldrich), and control siRNA (D-001810-10-05, Dharmacon).

For experiments in U2OS cells expressing different CENP-E variants, endogenous CENP-E was depleted by transfecting cells with 20 nM 3′UTR-targeting siRNA (5′-CCACUAGAGUUGAAAGAUA-3′) 24 h prior to fixation or imaging[43]. GFP-CENP-E expression was induced by adding doxycycline (1 μg/ml, Sigma-Aldrich) overnight. To label DNA, 1 nM SPY650-DNA (SC501, Spirochrome) was added 6 h before imaging together with the broad-spectrum efflux pump inhibitor verapamil (1 μM, Spirochrome). U2OS CENP-A-GFP-expressing cells were treated with CENP-E siRNA as described above for RPE-1 cells.

## Drug treatments and drug washouts

Aurora B inhibitor Barasertib (AZD1152-HQPA, Selleckchem, IC$_{50}$ value 0.32 nM) at a final concentration of 300 nM, was added acutely before imaging. CENP-E inhibitor GSK-923295 (MedChemExpress, IC50 value 3.2 nM) at a final concentration of 80 nM for RPE-1 and 200 nM for U2OS cells, to achieve the same effect, was added 1–4 h before imaging in most experiments, or when noted acutely before imaging. Eg5 inhibitor Monastrol (HY-101071A/CS-6183, MedChemExpress, IC$_{50}$ value 50 μM) at a final concentration of 100 μM, was added 3 h before imaging or acutely before imaging when noted. Aurora B inhibitor ZM-447439 (MedChemExpress, IC$_{50}$ value 130 nM) at a final concentration of 2 or 3 μM as noted, was added acutely before live imaging or 15 min before fixation as noted. Aurora A inhibitor MLN8237

(MedChemExpress, IC$_{50}$ value 4 nM) at a final concentration of 125 nM or 250 nM as noted, was added acutely before imaging or 30–60 min before fixation. Bub1 inhibitor BAY-320 (MedChemExpress, IC$_{50}$ value 680 nM) at a final concentration of 10 μM, was added acutely before imaging. Haspin inhibitor 5-Iodotubercidin (5-ITu) (MedChemExpress, IC$_{50}$ value 5–9 nM) at a final concentration of 2 μM, was added acutely before imaging. Mps1 inhibitor AZ3146 (MedChemExpress, USA, IC$_{50}$ value 35 nM) at a final concentration of 500 nM, was added acutely before imaging. PLK1 inhibitor BI 2536 (MedChemExpress, IC$_{50}$ value 0.83 nM) at a final concentration of 100 nM, was added acutely before imaging. Inhibitor of PP1 and PP2A Okadaic acid (MedChemExpress, IC$_{50}$ value 0.1–0.3 nM for PP2A and 15–50 nM for PP1) at a final concentration of 1 μM, was added acutely before imaging or 30 min before washout of the CENP-E inhibitor as noted. Cytoplasmic dynein inhibitor Ciliobrevin D (HY-122632, MedChemExpress) at a final concentration of 20 μM, was added acutely before imaging. Proteasome inhibitor MG-132 (Merck, IC$_{50}$ value 100 nM) at a final concentration of 1 μM, was added acutely before imaging or together with CENP-E inhibitor as noted.

The stock solutions for all drugs were prepared in DMSO except Okadaic acid which was prepared in ethanol. The stock solutions of all drugs were kept aliquoted at 10–50 μL at −20 °C for a maximum period of three months or at −80 °C for a maximum period of six months. New aliquots were thawed weekly for each new experiment. All drugs were added to DMEM used for cell culture to obtain the final concentration of a drug as described, except Ciliobrevin D, which was added to a serum-reduced Opti-MEM that was placed on cells acutely before adding the drug, as Ciliobrevin D did not show any effect when used in DMEM with 10% FBS[75]. Drug washouts were performed by replacing drug-containing medium with 2 mL of fresh pre-warmed DMEM followed by four subsequent washouts with 2 mL of pre-warmed DMEM.

Acute inhibition of Aurora B by ZM-447439 (2 or 3 μM) or barasertib (300 nM) induced premature anaphase onset 15–30 min after the addition of Aurora B inhibitor, as reported previously[34]. Similarly, premature onset of anaphase was observed during the 30 min course of the imaging in cells acutely treated with the Mps1 inhibitor AZ3146 (500 nM), as expected from previous work[57]. When noted in the text, we treated CENP-E inhibited cells with the proteasome inhibitor MG-132 together with Aurora B or Mps1 inhibitors to block premature anaphase onset. MG-132 treatment provided more time for the chromosomes to complete the alignment successfully, as in the Aurora B inhibitor alone, anaphase ensued while some chromosomes were still in the alignment process.

Acute treatment of cells with dynein inhibitor 20 μM Ciliobrevin D splayed the spindle poles in all treated cells during 30 min after the addition of the drug, reflecting the reported role of the dynein-dynactin complex in the focus of the spindle poles in human cells[75]. Interestingly, co-inhibition of Bub1 by BAY-320 (10 μM) and Haspin by 5-ITu (2 μM) in some cells induced premature onset of anaphase during the first 20 min after addition of the drug, reflecting the possible role of both pathways in the maintenance of the spindle assembly checkpoint[59]. The 30-min long Okadaic acid treatment (1 μM) led to depolymerization of actin filaments and rounding of interphase cells[100]. Acute treatment of cells with Okadaic acid (1 μM) together with acute treatment with Aurora B inhibitor or after CENP-E inhibitor washout resulted in frequent expels of aligned kinetochores from the plate and sometimes complete breakage of the integrity of metaphase plates, but only after most of the polar chromosomes had already aligned to the plate, similar to previous reports of PP1 antibody injections into HeLa cells[26]. After acute treatment of CENP-E inhibited cells with the Eg5 inhibitor monastrol (50 μM), spindles quickly shortened, as expected from previous reports[101]. Similarly, acute and drastic spindle shortening was observed in most CENP-E inhibited cells after acute addition of the Aurora A inhibitor MLN8237 (125 nM), as expected from previous studies[66]. Acute inhibition of PLK1 by BI 2536

(100 nM) induced a fast block of anaphase spindle elongation and cytokinetic furrowing in cells after CENP-E washout, as expected from previous reports[102].

Regarding cells imaged by confocal microscopy after the chronic inhibition of CENP-E by GSK-923295 (80 nM), all pseudo-metaphase spindles chosen for imaging were phenotypically similar between each other and between conditions where CENP-E was depleted or reactivated after washout of the CENP-E inhibitor: (1) in all cells few chromosomes were localized close to one of the spindle poles, called polar chromosomes, and their numbers ranged from 2 to 16 per cell; (2) occasionally in some cells few chromosomes were localized already in between spindle pole and equator; and (3) in all cells most chromosomes were already aligned and tightly packed at the metaphase plate. This type of chromosome arrangement was expected from previously published data that reported that only 10–30% of chromosomes upon NEBD require CENP-E-mediated alignment[4]. Likewise, under all conditions where CENP-E was perturbed and cells imaged by confocal microscopy during pseudo-metaphase, the overall displacement of spindle poles from each other was negligible, consistent with cells being in a pseudo-metaphase state where there is no net change in spindle length. For LLSM-based assay, no inclusion criteria for imaging were used as the cells were non-synchronized and entered mitosis stochastically. Only randomly selected cells that entered mitosis and subsequently entered anaphase during imaging time were used for kinetochore and centrosome tracking analysis.

## Immunofluorescence

All cells were fixed for 2 min with ice-cold methanol, except for those used in STED microscopy, which are described below. After fixation, cells were washed 3 times for 5 min with 1 ml of PBS and permeabilized with 0.5% Triton-X-100 solution in water for 30 min at room temperature. To block unspecific binding, cells were incubated in 1 ml of blocking buffer (2% bovine serum albumin, BSA, A7906, Sigma-Aldrich or 2% normal goat serum, NGS, 10000C, Thermo-Fisher) for 2 h at room temperature. The cells were then washed 3 times for 5 min with 1 ml of PBS and incubated with 500 µl of primary antibody solution overnight at 4 °C. Antibody incubation was performed using a blocking solution composed of 0.1% Triton, 1% BSA in PBS.

Following primary antibodies were used: rabbit monoclonal Anti-NDC80 (HPA066330-100 µL, Sigma-Aldrich, diluted 1:250), mouse monoclonal anti-KIFC1 (M-6, sc-100947, Santa Cruz, diluted 1:500), rabbit polyclonal anti-Kif18a (A301-080A, Bethyl Laboratories, diluted 1:500), mouse monoclonal anti-BubR1 (MAB3612, EMD Milipore, diluted 1:500), rabbit anti-Kif4A (A301-074A, Bethyl Laboratories, diluted 1:500), rabbit anti-SPINDLY/CCD98 (A301-354A, Bethyl Laboratories, diluted 1:500), mouse monoclonal anti-KID (B-9, sc-390640, Santa Cruz, diluted 1:500), rabbit polyclonal anti-ZW10 (ab21582, Abcam, 1:500), mouse monoclonal anti-Astrin, clone C-1 (MABN2487, Merck, 1:250), and human anti-centromere (CREST) protein (15-234, Antibodies Incorporated, 1:500). Where indicated, DAPI (1 µg/mL) was used for chromosome visualization.

After primary antibody, cells were washed in PBS and then incubated in 500 µL of secondary antibody solution for 1 h at room temperature. Following secondary antibodies were used: Donkey anti-rabbit IgG Alexa Fluor 594 (ab150064, Abcam, diluted 1:1000) for all rabbit antibodies, Donkey anti-mouse IgG Alexa Fluor 594 (ab150108, Abcam, diluted, 1:1000) for all mouse antibodies except anti-KID which was conjugated with Donkey anti-mouse IgG Alexa Fluor 647 (ab150107, Abcam, diluted 1:1000), goat anti-human DyLight 594 (Abcam, ab96909, diluted 1:1000), and Donkey anti-rat IgG Alexa Fluor 647 (ab150155, Abcam, diluted 1:500). Finally, cells were washed with 1 ml of PBS, 3 times for 10 min. Cells were imaged either immediately following the imaging or were kept at 4 °C before imaging for a maximum period of one week.

To visualize alpha-tubulin in STED resolution in the RPE-1 CENP-A-GFP centrin1-GFP cell line, the ice-cold methanol protocol was avoided because it destroyed the unstable fraction of microtubules[103]. Cells were washed with cell extraction buffer (CEB) and fixed with 3.2% paraformaldehyde (PFA) and 0.1% glutaraldehyde (GA) in PEM buffer (0.1 M PIPES, 0.001 M MgCl2 × 6 H2O, 0.001 M EGTA, 0.5% Triton-X-100) for 10 min at room temperature. After fixation with PFA and GA, for quenching, cells were incubated in 1 ml of freshly prepared 0.1% borohydride in PBS for 7 min and then in 1 mL of 100 mM $NH_4Cl$ and 100 mM glycine in PBS for 10 min at room temperature. To block nonspecific binding of antibodies, cells were incubated in 500 µL blocking/permeabilization buffer (2% normal goat serum and 0.5% Triton-X-100 in water) for 2 h at room temperature. Cells were then incubated in 500 µL of primary antibody solution containing rat anti-alpha-tubulin YL1/2 (MA1-80017, Invitrogen, diluted 1:500) overnight at 4 °C. After incubation with a primary antibody, cells were washed 3 times for 10 min with 1 ml of PBS and then incubated with 500 µl of secondary antibody containing donkey anti-rat IgG Alexa Fluor 594 (ab150156, Abcam, diluted 1:300) for 2 h at room temperature, followed by wash with 1 ml of PBS 3 times for 10 min.

## Imaging

Opterra I multipoint scanning confocal microscope system (Bruker Nano Surfaces) was mainly used for live-cell imaging of hTERT-RPE-1 cells expressing CENP-A-GFP and Centrin1-GFP, in cells depleted of CENP-E by siRNAs and in cells with inhibited CENP-E by CENP-E inhibitor, in the assay we termed a confocal-based imaging assay. The system was mounted on a Nikon Ti-E inverted microscope equipped with a Nikon CFI Plan Apo VC 100x/1.4 numerical aperture oil objective (Nikon). The system was controlled with Prairie View Imaging Software (Bruker). During imaging, cells were kept at 37 °C and at 5% $CO_2$ in the Okolab cage incubator (Okolab). The laser power was set to 10% for the 488 nm excitation laser. For optimal resolution and signal-to-noise ratio, a 60 µm pinhole aperture was used and the xy-pixel size was set to 83 nm. For excitation of GFP, a 488-nm diode laser line was used. The excitation light was separated from the emitted fluorescence using the Opterra dichroic and Barrier 488 eGFP filter set (Chroma). Images were acquired with an Evolve 512 Delta Electron Multiplying Charge Coupled Device (EMCCD) camera (Photometrics) using a 150-ms exposure time and with a camera readout mode of 20 MHz. 30-60 z-stacks encompassing all focal planes with visible green signal were acquired with 0.5 µm spacing to cover the entire spindle area in depth with unidirectional xyz scan mode and with the "fast acquisition" option enabled in software. Images of six different cells were taken simultaneously every 30s-1 min. The total duration of the time-lapse movies was 30 min to 1 h. Movies of control U2OS cells imaged on Opterra I microscope system cells were obtained from a previous study[8].

The STED confocal microscope system (Abberior Instruments) and the LSM800 laser scanning confocal system (Zeiss) were used for the remainder of the live-cell experiments we termed the confocal-based imaging assay and in hTERT-RPE-1 cells expressing CENP-A-mCerulean and Mad2-mRuby after reactivation of CENP-E activity. STED microscopy was also used to image all fixed cells in super-resolution. STED microscopy was performed using an Expert Line easy3D STED microscope system (Abberior Instruments) with the 100 x/1.4NA UPLSAPO100x oil objective (Olympus) and an avalanche photodiode detector (APD). The 488 nm line was used for excitation, with the addition of the 561 nm line for excitation and the 775 nm laser line for depletion during STED super-resolution imaging. The images were acquired using Imspector software. The xy pixel size for fixed cells was 20 nm and 10 focal planes were acquired with a 300 nm distance between the planes. For confocal live cell imaging of cells, the xy pixel size was 80 nm and 16 focal images were acquired, with 0.5 µm distance between the planes and 1 min time intervals between different

frames. During imaging, live cells were kept at 37 °C and at 5% $CO_2$ in the Okolab stage incubation chamber system (Okolab).

LSM 800 confocal laser scanning microscope system (Zeiss) was used for the rest of live-cell confocal-based imaging in hTERT-RPE-1 cells expressing CENP-A-GFP and centrin1-GFP, with the following parameters: sampling in xy, 0.27 μm; z step size, 0.5 μm; total number of slices, 32; pinhole, 48.9 μm; unidirectional scan speed, 10; averaging, 2; 63x Oil DIC f/ELYRA objective (1.4 NA), 488 nm laser line (0.1-1% power for different experiments), and detection ranges of 450–558 nm for the green channel, 561 nm laser line (0.1-1% power for different experiments) and detection range of 565–650 nm for the red channel, 640 nm laser line (0.1–1% power for different experiments), and detection range of 656–700 nm for the far red channel, and 405 nm laser line (0.5% power) and detection range of 400–450 for the blue channel. Images were acquired using ZEN 2.6 (blue edition; Zeiss). Six cells in volumes of 15 μm were imaged sequentially every 1 min at different positions determined right after the addition of a drug. During imaging, cells were incubated at 37 °C and 5% $CO_2$ using a Pecon stage incubation chamber system (Pecon, Heating Insert P S1, #130-800 005). The system was also used for imaging of all other fixed cells that were not imaged by STED microscopy, on multiple positions and at the confocal lateral resolution.

The Lattice Lightsheet 7 microscope system (Zeiss) was used for live cell imaging of hTERT-RPE-1 cells expressing CENP-A-GFP and Centrin1-GFP and U2OS cells expressing CENP-A-GFP after reactivation of CENP-E activity, in cells depleted of CENP-E by siRNAs in cells with inhibited CENP-E by CENP-E inhibitor, and in U2OS cells expressing CENP-E mutants, in the assay we called LLSM-based imaging assay. The system was equipped with an illumination objective lens 13.3×/0.4 (at a 30° angle to cover the glass) with a static phase element and a detection objective lens 44.83×/1.0 (at a 60° angle to cover the glass) with an Alvarez manipulator. Images were acquired using ZEN 2.6 (blue edition; Zeiss). The automatic immersion of water was applied from the motorized dispenser at an interval of 20 or 30 min. Right after sample mounting, four steps of the 'create immersion' auto immersion option were applied. The sample was illuminated with a 488-nm diode laser (power output 10 Mw) with laser power set to 1–2%. The detection module consisted of a Hamamatsu ORCA-Fusion sCMOS camera with exposure time set to 15–20 ms. The LBF 405/488/561/642 emission filter was used. During imaging, cells were kept at 37 °C and at 5% $CO_2$ in a Zeiss stage incubation chamber system (Zeiss). The width of the imaging area in the x dimension was set from 1 to 1.5 mm, with a 0.4 μm interval size. The time between consecutive frames varied from 30 seconds to 1 minute, depending on the chosen imaging area width. The total imaging duration, set for 1 to 1.5 days, was occasionally interrupted by air bubbles, which caused a loss of intensity in part or all of the imaging area. When the entire area was affected by air bubbles, the image was cropped in ZEN software to reduce the final file size before further processing. Some movies, which were later processed for color-coding, were deskewed using ZEN 3.7 software with the "Linear Interpolation" and "Cover Glass Transformation" settings. The light sheet's length, also referred to as the field of view or illumination width, was 30 μm, while its thickness was set to 1000 nm. The parameters 'Focus sheet,' 'Focus Waist,' and 'Aberration Control' were manually fine-tuned before each imaging session, with ranges of -0.170 to -0.230, 50 to 60, and 170 to 185, respectively.

### Tracking of centrosomes and kinetochores
The spatial x and y coordinates of the kinetochore pairs were extracted in each time frame using the Low Light Tracking Tool (v.0.10), an ImageJ plugin based on the Nested Maximum Likelihood Algorithm (https://imagej.net/plugins/low-light-tracking-tool), as previously described[101]. Tracking of polar kinetochores in the x and y planes was performed on the maximum intensity projections of all acquired z planes. Some kinetochore pairs could not be successfully

tracked in all frames, mainly owing to cell and spindle movements in the z-direction over time. Spindle poles were manually tracked with points placed between the center of the two centrioles or centriole in centrinone treated cells. Kinetochore pairs that were aligned at the start of the imaging in a confocal-based imaging assay were also manually tracked in two dimensions. Quantitative analysis of all parameters was performed using custom-made MATLAB scripts (MatlabR2021a 9.10.0) scripts.

Due to the inability to reliably track polar kinetochores across all time frames due to neighboring kinetochores, tracking of polar kinetochores routinely commenced a few frames before the kinetochore pair began moving towards the equatorial plane. In the CENP-E reactivation assay, kinetochores were tracked from the onset of imaging until the successful identification of the same kinetochore pair, which occurred when the kinetochore pair was lost in the imaging plane, among other kinetochores, or when the imaging session ended, whichever came first. Aligned kinetochore pairs were tracked in the confocal-based assay using the same protocol.

### Quantification of mean Mad2 intensity of kinetochore pairs
The fluorescence intensity signal of each kinetochore in both CENP-A and Mad2 was measured by using the "*Oval selection*" tool in ImageJ with the size and position defined by the borders of the CENP-A signal of each kinetochore in the sum intensity projection of all acquired z-planes. The background fluorescence intensity measured in the cytoplasm was subtracted from the obtained values, and the calculated integrated intensity value was divided by the number of z-stacks used to generate the sum projection of each cell. The obtained mean intensity value subtracted for background for each kinetochore was normalized to the intensity of the CENP-A signal.

### Quantification of mean signal intensities of proteins after RNAi interreference
The fluorescence intensity signals were measured in triplicate samples for targeting and non-targeting groups for each siRNA treatment. Each siRNA targeting and non-targeting triplicates were imaged by the same protocol. All proteins were measured in mitotic cells at their respective locations during mitosis by using the "*Polygon selection tool*", as established by many previous studies: kinetochores and chromosomes defined by the DAPI or CENP-A area on sum projection for CENP-E, Spindly, BubR1, and on single planes for Hec1 defined by the CENP-A signal; entire spindle region defined by extent of centrin-1 and CENP-A signals on sum projections for Kif4a, Kif18a, and HSET; and chromosomes defined by DAPI signal on sum projections for Kid.

### Quantification of polar kinetochore pairs
Polar kinetochores in all experiments were defined as kinetochore pairs that are closer to one of the spindle poles than to the equatorial plane. In confocal-based experiments polar chromosomes were defined from the onset of imaging. The number of polar kinetochore pairs in cells that were imaged from the onset of mitosis, mainly related to LLSM-based experiments, was quantified 12 min from the onset of mitosis and then every 6 min until the onset of anaphase or until the end of the imaging on maximum intensity projections of all acquired z planes.

To calculate the distance between kinetochores in fixed samples, two points were placed at the center of the signal in each pair of kinetochores using a 'Point tool' in ImageJ. Congression velocity was calculated for each pair of kinetochores in the last 6 min before the center of the pair of kinetochores surpassed 2 μm from the equatorial plane measured as the nearest distance from a center of a pair to a plane. These times represent the fast kinetochore movement towards the plate. Aligned kinetochore pair was defined as every pair that was found 3 μm from the equatorial plane at any given time. The equatorial or metaphase plane was defined in each time frame as the line

perpendicular to the line connecting the centrosomes at their midpoint. The angle between the kinetochore pairs and the main spindle axis was defined as an angle between a line connecting two centrosomes and a line connecting the center of the kinetochore pair and a spindle pole nearest to the respective pair.

In images acquired using STED microscopy, lateral attachments were defined as those in which the kinetochore did not form visible attachments with microtubules that terminate at the kinetochore, defined by the CENP-A signal; end-on attachments were defined as those in which the kinetochore formed visible attachments with microtubules that terminated at the kinetochore, defined by the CENP-A signal, with no visible microtubule signal just below or above the kinetochore. When scoring successful alignment of polar chromosomes, alignment was considered successful if a chromosome, initially polar at the beginning of imaging, moved to within 2 μm of the equatorial plane after 30 min. The 30-min period was chosen because it represents the minimum imaging time used for the confocal-based microscopy assay, and it covers the majority of the imaged cells.

The alignment categories were defined as follows: "No metaphase plate" indicated that no metaphase plate is formed. "High misalignment" indicated a metaphase plate is formed, but more than 5 chromosomes are outside the plate. "Low misalignment" indicated a situation where a metaphase plate is formed, with fewer than 5 chromosomes outside the plate. Finally, "Tight metaphase plate" signified the formation of a tight metaphase plate with no chromosome misalignment. "Maximum spread of chromosomes" was measured on maximum intensity projections as the nearest distance between centers of two most separated kinetochore pairs.

### Image processing and statistical analysis

Image processing was performed in ImageJ (National Institutes of Health). Quantification and statistical analysis were performed in MatLab. The figures were assembled in Adobe Illustrator (Adobe Systems). Raw images were used for quantification. The images of spindles were rotated in every frame to fit the long axis of the spindle to be parallel with the central long axis of the box in ImageJ and the spindle short axis to be parallel with the central short axis of the designated box in ImageJ. The designated box sizes were cut to the same dimensions for all panels in the figures where the same experimental setups were used across the treatments. When comparing different treatments in channels in which the same protein was labeled, the minimum and maximum intensity of that channel was set to the values in the control treatment. When indicated, the smoothing of the images was performed using the "Gaussian blur" function in ImageJ ($s = 0.5–1.0$). Color-coded maximum intensity projections of the z-stacks were done using the *Temporal color code* tool in Fiji by applying "16-color" or "Spectrum" lookup-table (LUT) or other LUT as indicated. For the generation of univariate scatter plots, the open Matlab extension "UnivarScatter" was used (https://github.com/manulera/UnivarScatter).

Data are given as mean ± standard deviation (s.t.d.), unless otherwise stated. The mean line was plotted to encompass a minimum of 60% of the data points for each treatment. Other dispersion measures used are defined in their respective figure captions. The exact values of n are given in the respective figure captions, where n represents the number of cells and the number of tracked kinetochore pairs, as defined for each n in the figure captions or tables. The number of independent biological replicates is also given in the figure captions. An independent experiment for acute drug treatments was defined by the separate addition of a drug to a population of cells in a dish. The number of cells imaged simultaneously ranged from 1 to 7, depending on the specific microscopy system used. The p values when comparing data from multiple classes that followed a normal distribution were obtained using the one-way analysis of variance (ANOVA) test followed by pairwise Two-sided Tukey's Honest Significant Difference (HSD)

test (significance level was 5%). Group proportions were compared using two-tailed two-proportion z-tests, with overall differences assessed by a chi-square test of independence. $p < 0.05$ was considered statistically significant, very significant if $0.001 < p < 0.01$ and extremely significant if $p < 0.001$. Values of all significant differences are given with the degree of significance indicated ($*0.01 < p < 0.05$, $**0.001 < p < 0.01$, $***p < 0.001$, $**** < 0.0001$). Linear regression was performed using ordinary least squares. The significance of the slope was assessed with a two-tailed $t$ test, and goodness-of-fit was evaluated using $R^2$. The exact $p$ values are given in the Source data file[104]. No data were excluded from the analyses, unless otherwise stated. No statistical methods were used to predetermine the sample size. The experiments were not randomized and, except where stated, the investigators were not blinded to allocation during experiments and outcome evaluation.

### Reporting summary
Further information on research design is available in the Nature Portfolio Reporting Summary linked to this article.

## Data availability
The source data used for the main figures have been deposited in the Figshare database (https://doi.org/10.6084/m9.figshare.29086730)[104]. All other data supporting the findings of this study are available, and access can be obtained from the corresponding authors.

## Code availability
Codes used to analyze and plot the data are available from the corresponding authors on request.

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

## Acknowledgements

We thank Alexey Khodjakov, Jonathon Pines and Marin Barišić for cell lines; Marin Barišić, Carlos Conde, Helder Maiato and Andrea Musacchio for reviewing and discussing the manuscript; Julie Welburn for discussions and antibodies; Magda Topić and Mia Crnogaj for help with cell culture work and preparation of inhibitors; Ivana Šarić for the drawings; and members of the Tolić group and Nenad Pavin group for constructive comments on the manuscript. This work was funded by the European Research Council (ERC Synergy Grant, GA Number 855158), the Croatian Science Foundation (HRZZ) through Swiss-Croatian Bilateral Projects (project IPCH-2022-10-9344), and projects co-financed by the Croatian Government and the European Union through the European Regional Development Fund—the Competitiveness and Cohesion Operational Program: IPSted (Grant KK.01.1.1.04.0057) and QuantiXLie Center of Excellence (Grant KK.01.1.1.01.0004). This research was performed using services, storage and computing resources provided by the University of Zagreb University Computing Center – SRCE.

## Author contributions

K.V. and I.M.T. conceived the project, K.V. performed all experiments, quantified, analyzed and presented the data, K.V. conceptualized and prepared original draft, K.V. and I.M.T. reviewed, edited and discussed the manuscript.

## Competing interests

The authors declare no competing interests.
