## [Transparent Peer Review file · Nature Communications]

CENP-E initiates chromosome congression by opposing Aurora kinases to promote end-on attachments

Corresponding Author: Professor Iva Tolić

Version 0:

Reviewer comments:

Reviewer #1

(Remarks to the Author)

Chromosome congression to the spindle equator is crucial for accurate cell division. Mitotic kinesin CENP-E is uniquely required for the rapid congression of chromosomes to the equator. Current models suggest that CENP-E drives congression by gliding kinetochores along microtubules independently of chromosome biorientation. Vukušić and Tolić propose an alternative model in which CENP-E initiates congression by promoting the formation or stabilization of end-on attachments at kinetochores, rather than by directly propelling the chromosomes by studying chromosome movement using a lattice light sheet microscope with different levels of CENP-E activity. They propose that CENP-E counters an inhibition by reducing Aurora B-mediated phosphorylation of outer kinetochore proteins in a BubR1-dependent manner which stabilizes end-on attachments by facilitating the removal of the fibrous corona and initiates biorientation-dependent chromosome movement. The work is well-performed, and definitely needs to be distributed to the field, while the following points require clarification.

Major Points:

1. Most of functional analyses were based on a combination of siRNA and chemical inhibitors which do not give sufficient time resolution to delineate mitotic chromosome movements discussed in this study. I would recommend authors to employ dTAG or AID system to perform time-resolved degradation of CENP-E to validate the chemical inhibitor experiments. Otherwise, the depletion condition by siRNA can not be confirmed.
2. As a control for GSK923295, a structure distinct compound with ability to inhibit CENP-E should be included. This includes CENP-E inhibitor syntelin (PMID: 21119683).
3. End-on capture should be validated by TEM.

Minor Points:

There are a few typos throughout manuscript. For example, Zw10 should be ZW10.

Reviewer #2

(Remarks to the Author)

The manuscript presents several significant findings that advance our understanding of chromosome congression during mitosis: 1) CENP-E's Role in Congression Initiation: The authors demonstrate that CENP-E is crucial for the initiation of chromosome congression by promoting the formation or stabilization of end-on attachments at kinetochores, rather than directly driving chromosome movement. This challenges the prevailing model that CENP-E propels chromosomes independently of biorientation. 2) Aurora Kinase Opposition: The study reveals that CENP-E accelerates congression initiation by opposing Aurora kinase activity, particularly Aurora B, which destabilizes end-on attachments. This finding provides a mechanistic link between CENP-E and Aurora kinases in regulating chromosome alignment. 3) Fibrous Corona Expansion: The authors show that in the absence of CENP-E, the fibrous corona expands, inhibiting end-on attachments and delaying congression. This highlights the role of the fibrous corona as a regulatory structure in chromosome alignment. 4) BubR1 Dependency: The study identifies BubR1 as a critical factor for CENP-E-mediated congression initiation, independent of its role in CENP-E recruitment. This adds a new layer of complexity to the regulatory network controlling

chromosome congression. These results are supported by a combination of live-cell imaging, super-resolution microscopy, and chemical inhibition experiments, providing robust evidence for the proposed model.

This work is of high significance to the field of cell biology, particularly in the context of mitosis and chromosome segregation. The findings challenge existing models of chromosome congression and provide a more nuanced understanding of how CENP-E, Aurora kinases, and BubR1 coordinate to ensure accurate chromosome alignment. The study also has implications for related fields, such as cancer biology, where misregulation of chromosome congression can lead to aneuploidy and genomic instability.

The manuscript builds on and challenges previous models of chromosome congression. While earlier studies suggested that CENP-E drives chromosome movement independently of biorientation, this work provides evidence that CENP-E's primary role is in initiating congression by stabilizing end-on attachments. This aligns with some recent studies that have hinted at the importance of end-on attachments in chromosome alignment but provides a more comprehensive mechanistic framework. The findings also complement previous work on Aurora kinases and BubR1, extending their roles to the initiation phase of congression.

In summary, this is a well-executed study that provides significant insights into the mechanisms of chromosome congression. The findings challenge existing models and propose a new framework for understanding how CENP-E, Aurora kinases, and BubR1 coordinate to ensure accurate chromosome alignment. The overall quality of the work is high, and the manuscript is suitable for publication after revisions.

- 1) It would be better if the authors could discuss how bioriented polar chromosomes move to the equatorial region of the spindle;
- 2) It would be better if the authors could address how CENP-E opposes the kinase activity of centrosomal Aurora A and centromere/kinetochore-localized Aurora B?
- 3) It would be better if the authors could test whether distinct pools of CENP-E on the kinetochore and fibrous corona promote the end-on attachment and drive the movement of polar chromosomes, respectively?
- 4) If I understand correctly, in the working model showing in Figure 8B, the blunt line from Aurora B to Knl1/Ndc80 should be to the arrow line between Knl1/Ndc80 and End-on attachment.
- 5) In lines 376-378, the authors mentioned that "On the other hand, inhibition and co-inhibition of Bub1 (Budding uninhibited by benzimidazoles or Haspin, factors essential for the recruitment of Aurora B to the outer and inner centromere, respectively 61–64". The following literatures should be cited as well: Wang F et al., Science 2010 (PMID: 20705812); Yamagishi Y et al., Science 2020 (PMID: 20929775); Liang C et al., J Cell Biol 2020 (PMID: 31868888).
- 6) In lines 382-384, the authors mentioned that "the control group was Monopolar spindle 1 (Mps1) kinase (Fig. 5C; and Fig. S4N). This implies that centrosome localized Aurora A and the outer-kinetochore localized Aurora B are major factors that limit the initiation of congression in the absence of CENP-E". In addition, in lines 574-576, the authors mentioned that "Polar chromosomes initially exhibit lateral attachments of kinetochores to microtubules, which fail to quickly convert to stable end-on attachments without CENP-E due to the high activity of outer-kinetochore localized Aurora B (AurB)". However, the authors did not provide evidence showing that it is the outer-kinetochore localized Aurora B that executes the function.

Reviewer #3

(Remarks to the Author)

Bioriented kinetochore-microtubule attachments are essential for the accurate segregation of chromosomes. The authors explore how CENP-E initiates congression and promotes end-on attachments to ensure proper biorientation. In previous studies, it has been shown that CENP-E is essential for end-on attachments. Here using a combination of gentle high-resolution live-cell microscopy and CENP-E inhibitor treatment the authors show that the kinesin is required to form or stabilize end-on attachments rather than directly powering the chromosome movement. This is likely because CENP-E accelerates congression initiation by opposing Aurora kinase activity, but chromosome movement toward the equator does not require CENP-E. Also stabilization of end-on attachments is delayed in the absence of CENP-E due to Aurora kinase-mediated phosphorylation of microtubule-binding proteins on kinetochores and the expansion of the fibrous corona. The authors speculate that CENP-E reduces Aurora B-mediated phosphorylation of outer kinetochore proteins in a BubR1-dependent manner and thereby stabilizes end-on attachments which is necessary for the full removal of the fibrous corona and biorientation.

Major comments.

Figure 2 shows change in Mad2 levels associated with lateral to end-on conversion. Can the authors compare levels in poleward vs non-poleward movement of sisters to indicate whether one of the sisters is more end-on tethered than the other? This is important because the conclusions are currently made using one marker Mad2 which has dynein-dependent and independent pathways to leave the kinetochore.

Kuhn and Dumont 2017 use SKAP as a marker to distinguish end-on attachment from lateral attachment based long-lived force. Having a second marker beyond Mad2 can help some of the key conclusions.

Lateral microtubule attachments were found in EM studies of congressed kinetochores (Magidson et al., 2011). Here using STED the authors show that congressing chromosomes are bioriented a few microns away from centrosomes. The former was an observation made in unperturbed cells while this study focusses on chromosomes positioned near the poles. The authors may benefit discussing these two cases, instead of ruling out one or the other. In fact the authors do find that Aurora kinases (at poles) can delay congression in the absence of CENP-E activity. Moreover, it will help explain why only some but not all kinetochores remain uncongressed in CENP-E inhibitor treated cells.

Would it help the authors to exploit STLC or monastrol release for studying biorientation of kinetochores that are randomly near the equator (far away from poles) but attempting to align at kinetochores? Or can they compare congression efforts (persistent travel or Inter-kinetochore distances) close to the equator versus pole using the data they have?

Minor comments?

Figure 1A – why does the 1.5mm line span the magnified crop with a 5micron scale bar?
Please include immunoblot or immunofluorescence to show extent of CENP-E depletion.

Figure S3A and B: typo 'reactivated'

In figure 6E – is there a lowering of intensity with time?

Figure 7 interpretation is somewhat confusing. In HEC1-depleted cells end-on attachments can not form, so we do expect that CENP-E;HEC1 codepletion to present severe misalignment.

Also HEC1 depletion causes a prolonged mitotic arrest, and during this arrest kinetochores often worsen their congression status making it difficult to build a direct correlation between congression status and HEC1 levels. Live-cell imaging and then a fixation at the end of the tracking may be helpful here for clear conclusions.

Please clarify these sentences:

“This suggests that issues with error correction, caused by Aurora B inhibition⁵⁰, limit the complete congression of some polar chromosomes.”

In figure legend 8 “Functional negative feedback loop of signaling interactions”

Version 1:

Reviewer comments:

Reviewer #1

(Remarks to the Author)

Thank you for addressing all my concerns. The manuscript has been greatly improved. I have no further concerns.

Reviewer #2

(Remarks to the Author)

The authors have addressed all my comments. I therefore support the publication of this important study.

Reviewer #3

(Remarks to the Author)

In the manuscript titled “CENP-E initiates chromosome congression by opposing Aurora kinases to promote end-on attachments”. The authors have addressed all my queries satisfactorily. Recommendations below are largely clarifications on literature (citations) and data analysis.

1) In figure 3J, ‘congression’ and ‘alignment’ are somewhat confusing as it seems like aligned chromosomes have to be at equator whereas congressed is based on movement. Would it help them to call it ‘congressing’ and ‘aligned’ based on the precise phenotype? Or explain in results text how these phenotypes were separated in time.

2) The above comment is important in the context of abstract statement “These findings support a unified 2D model of chromosome movement in which congression is intrinsically coupled to biorientation.” Does the statement indicated that alignment is not coupled to biorientation, and only congression is coupled to biorientation? Some text clarification or abstract revision would be helpful for clarity.

3) Citation: Original papers missed.

3.1 Consider discussing your findings in the context of CENPE siRNA and inhibitor work from PMC6080938

3.2 Revise references related to “expressing the SAC protein Mad2, which accumulates on kinetochores lacking stable end-on attachments^{44–46}”. See figure 4 of PMID: 26258631 on correlative LM/EM status for Mad2.

3.3 Revise references related to “Astrin, a positive marker of stable end-on attachments^{47,48}”: Astrin-SKAP was established as proteins at the kinetochore in the papers cited (Ref 47 and 48) and the Astrin-SKAP complex was demonstrated as an end-on attachment marker in PMC3748344 and PMC5461026.

4) In figure 8 model, if CENPE initiates end-on attachments, how are bioriented kinetochores seen in CENPEinhibitor treated cells. Should it be ‘CENPE initiates end-on attachments of polar chromosomes’? Some specification in legend and cartoon would help clarity.

Rebuttal letter Vukušić&Tolić, NCOMMS-24-76214

We are extremely grateful to the reviewers for their thorough analysis of our manuscript, as well as for their thoughtful critique and encouraging comments. Point-by-point responses to each comment are provided below. All new or revised text in the manuscript is marked in blue.

Reviewer #1 (Remarks to the Author):

Chromosome congression to the spindle equator is crucial for accurate cell division. Mitotic kinesin CENP-E is uniquely required for the rapid congression of chromosomes to the equator. Current models suggest that CENP-E drives congression by gliding kinetochores along microtubules independently of chromosome biorientation. Vukušić and Tolić propose an alternative model in which CENP-E initiates congression by promoting the formation or stabilization of end-on attachments at kinetochores, rather than by directly propelling the chromosomes by studying chromosome movement using a lattice light sheet microscope with different levels of CENP-E activity. They propose that CENP-E counters an inhibition by reducing Aurora B-mediated phosphorylation of outer kinetochore proteins in a BubR1-dependent manner which stabilizes end-on attachments by facilitating the removal of the fibrous corona and initiates biorientation-dependent chromosome movement. The work is well-performed, and definitely needs to be distributed to the field, while the following points require clarification.

Major Points:

The reviewer: 1. Most of functional analyses were based on a combination of siRNA and chemical inhibitors which do not give sufficient time resolution to delineate mitotic chromosome movements discussed in this study. I would recommend authors to employ dTAG or AID system to perform time-resolved degradation of CENP-E to validate the chemical inhibitor experiments. Otherwise, the depletion condition by siRNA can not be confirmed.

The authors: We thank the reviewer for this constructive comment. In our view, the use of CENP-E inhibitors represents the fastest currently available method to perturb CENP-E activity, allowing for the study of its acute effects on mitotic chromosome movements in comparison to AID system, which requires tens of minutes to hours to deplete proteins. However, we agree that an alternative approach based on a completely different technology that retains most of the CENP-E protein intact but still perturbs its activity could provide valuable complementary insight. To this end, we utilized stable U2OS cell lines with doxycycline-inducible expression of GFP-tagged wild-type (WT) CENP-E or a phospho-null CENP-E mutant in which the Aurora A/B-specific phosphorylation site Threonine 422 is mutated (T422A) (obtained from Eibes et al., 2024, Nat Commun). This is now incorporated into Results:

“To quantify the dynamics of chromosome congression initiation in the absence of CENP-E motor activity but without directly perturbing the motor domain, we used an osteosarcoma U2OS cell line with doxycycline-inducible expression of a GFP-tagged phospho-null T422A mutant (Fig. 1c), which blocks CENP-E phosphorylation by Aurora kinases^{26,43}. We compared the timing of congression initiation in these cells to that in cells expressing inducible GFP-tagged wild-type (WT) CENP-E and in CENP-E-depleted cells (Fig. 1c, Extended data Fig. 3a). Quantification of congression events over time revealed that T422A-expressing cells initiated congression of polar chromosomes similarly to CENP-E-depleted cells, whereas WT CENP-E expression enhanced congression initiation efficiency (Fig. 1d). These findings suggest that the T422A mutant functionally mimics CENP-E loss⁴³, supporting the conclusion that polar chromosome initiation can proceed independently of CENP-E activity.

To examine kinetochore dynamics during congression in transformed cells, we imaged U2OS cells expressing CENP-A-GFP and measured congression velocities under untreated, CENP-E-depleted, and CENP-E-inhibited conditions (Fig. 1c, Extended data Fig. 3b, c). Consistent with our previous findings in RPE-1 cells (Fig. 1e, Extended Data Fig. 1b), both congression velocity (Fig. 1e, Extended Data Fig. 3d) and the extent of interkinetochore stretch (Extended Data Fig. 3e) during the final four minutes before alignment were similar across conditions with varying activity of CENP-E. Together, these results demonstrate that while CENP-E is not moving chromosomes during congression, it is essential for the timely initiation of congression in both transformed and non-transformed cell types (Fig. 1f).”

The reviewer: 2. As a control for GSK923295, a structure distinct compound with ability to inhibit CENP-E should be included. This includes CENP-E inhibitor syntelin (PMID: 21119683).

The authors: We thank the reviewer for this comment and agree that it would be ideal to confirm our findings using a different CENP-E inhibitor. We attempted to test syntelin, the only commercially available alternative we could identify (purchased from MedChemExpress). However, in our hands, this commercial syntelin had no discernible effect on chromosome dynamics.

We also tried to obtain syntelin directly from the laboratory that developed the compound and has published several studies using it (Dr. XueBiao Yao's group). Unfortunately, due to chemical import regulations from China, the shipping and customs process exceeded the reasonable time frame available for this manuscript revision.

Nevertheless, we now include in the Results the notion that the return of a subset of polar chromosomes over time can also be observed in published images of syntelin-treated HeLa cells (e.g., 10.1038/cr.2010.167 and 10.1093/jmcb/mjz051). This supports the concept that syntelin likely produces a similar effect on polar chromosome resolution as the CENP-E ATPase inhibitor used in our study, and as the CENP-E depletion.

“How do the dynamics of chromosome congression compare between cells with the active versus perturbed CENP-E? Under CENP-E depletion or inhibition, the number of polar chromosomes gradually decreased over time as chromosomes congressed toward the metaphase plate (Fig 1d; and Video 3). Despite the absence of CENP-E or its activity, the majority of polar chromosomes successfully congressed within 3 hours, consistent with previous observations using CENP-E depletion or distinct small-molecule inhibitors^{4,26,34,40-42}.”

The reviewer: 3. End-on capture should be validated by TEM.

The authors: We thank the reviewer for this important suggestion. While we agree that transmission electron microscopy (TEM) is the gold standard for identifying end-on kinetochore-microtubule attachments, we think that this approach, although informative, have significant limitations for studying chromosome congression, especially during early mitosis or in the context of CENP-E-independent mechanisms.

First, capturing a sufficient number of chromosomes during the dynamic process of congression by TEM is exceedingly challenging, even with enrichment protocols. This limitation is well

illustrated in a prior study (Kapoor et al., 2006, Science), where only a one mitotic cell was imaged, and end-on attachment was observed on only one sister kinetochore of a congressing chromosome. Second, the status of end-on attachments on moving chromosomes may be transient due to the low number of stable microtubule connections during early congression, making it difficult to draw broad conclusions from static images of a small number of cells.

To address these limitations (see also comment 1 by reviewer 3) and to study end-on attachment in a more comprehensive and high-throughput manner, we performed a large-scale immunofluorescence-based assay under four conditions: (1) cells continuously treated with the CENP-E inhibitor, (2) cells fixed at 8 minutes after inhibitor washout, (3) cells fixed at 15 minutes post-washout, and (4) DMSO-treated control cells. We stained for Astrin/SKAP complex, a known marker of stable end-on attachment, and CENP-E, a fibrous corona marker. We quantified the mean intensity of Astrin and CENP-E at every polar kinetochore and at randomly selected aligned kinetochores across these conditions. Additionally, we measured the distance of each kinetochore from the nearest spindle pole. This is now incorporated into Results:

“To independently test whether stabilization and maturation of end-on microtubule attachments occur at congressing chromosomes, we stained RPE-1 cells for Astrin, a positive marker of stable end-on attachments^{47,48} and CENP-E, a marker of fibrous corona^{30,49}. To capture the full range of kinetochore positions from the spindle poles to the equatorial plate, we analyzed cells under CENP-E inhibition, at two time points following CENP-E reactivation, and in DMSO-treated prometaphase cells undergoing congression (Fig. 3f). We quantified the intensity of Astrin and CENP-E at all polar kinetochores and at randomly selected aligned kinetochores, together with their distance from the nearest spindle pole. Following CENP-E reactivation, Astrin levels at polar kinetochores progressively increased as kinetochores were found closer to equatorial plane, reaching values comparable to those at aligned kinetochores (Fig. 3g). Increase in Astrin on congressing kinetochores mirrored both Mad2 loss (Fig. 2f) and end-on attachment maturation observed by STED microscopy (Fig. 3d) as the distance from the pole increased. Notably, by 15 minutes post-washout, Astrin levels at congressing kinetochores became indistinguishable from those in DMSO-treated prometaphase cells (Fig. 3g). In contrast, CENP-E levels declined as kinetochores moved toward the metaphase plate across conditions (Fig. 3h). CENP-E and Astrin display a mutually exclusive localization pattern at kinetochores: kinetochores enriched in CENP-E exhibit low Astrin levels, whereas those with high Astrin show reduced CENP-E presence (Fig. 3i). These results support a model in which chromosome congression involves the progressive capture and stabilization of end-on microtubule attachments (Fig. 3j), both following CENP-E reactivation and during unperturbed mitosis. Overall, congressing chromosomes are biochemically characterized by intermediate levels of checkpoint proteins (e.g., Mad2), intermediate Astrin levels indicating the maturation of end-on attachments, and reduced levels of fibrous corona proteins like CENP-E, which reflects ongoing corona stripping.”

Minor Points:

There are a few typos throughout manuscript. For example, Zw10 should be ZW10.

The authors: We have corrected this.

Reviewer #2 (Remarks to the Author):

The manuscript presents several significant findings that advance our understanding of chromosome congression during mitosis: 1) CENP-E's Role in Congression Initiation: The authors demonstrate that CENP-E is crucial for the initiation of chromosome congression by promoting the formation or stabilization of end-on attachments at kinetochores, rather than directly driving chromosome movement. This challenges the prevailing model that CENP-E propels chromosomes independently of biorientation. 2) Aurora Kinase Opposition: The study reveals that CENP-E accelerates congression initiation by opposing Aurora kinase activity, particularly Aurora B, which destabilizes end-on attachments. This finding provides a mechanistic link between CENP-E and Aurora kinases in regulating chromosome alignment. 3) Fibrous Corona Expansion: The authors show that in the absence of CENP-E, the fibrous corona expands, inhibiting end-on attachments and delaying congression. This highlights the role of the fibrous corona as a regulatory structure in chromosome alignment. 4) BubR1 Dependency: The study identifies BubR1 as a critical factor for CENP-E-mediated congression initiation, independent of its role in CENP-E recruitment. This adds a new layer of complexity to the regulatory network controlling chromosome congression. These results are supported by a combination of live-cell imaging, super-resolution microscopy, and chemical inhibition experiments, providing robust evidence for the proposed model. This work is of high significance to the field of cell biology, particularly in the context of mitosis and chromosome segregation. The findings challenge existing models of chromosome congression and provide a more nuanced understanding of how CENP-E, Aurora kinases, and BubR1 coordinate to ensure accurate chromosome alignment. The study also has implications for related fields, such as cancer biology, where misregulation of chromosome congression can lead to aneuploidy and genomic instability.

The manuscript builds on and challenges previous models of chromosome congression. While earlier studies suggested that CENP-E drives chromosome movement independently of biorientation, this work provides evidence that CENP-E's primary role is in initiating congression by stabilizing end-on attachments. This aligns with some recent studies that have hinted at the importance of end-on attachments in chromosome alignment but provides a more comprehensive mechanistic framework. The findings also complement previous work on Aurora kinases and BubR1, extending their roles to the initiation phase of congression. In summary, this is a well-executed study that provides significant insights into the mechanisms of chromosome congression. The findings challenge existing models and propose a new framework for understanding how CENP-E, Aurora kinases, and BubR1 coordinate to ensure accurate chromosome alignment. The overall quality of the work is high, and the manuscript is suitable for publication after revisions.

The Reviewer: 1) It would be better if the authors could discuss how bioriented polar chromosomes move to the equatorial region of the spindle;

The authors: We thank the reviewer for this comment. We now discuss in the Discussion what could be the mechanism that determines direction of movement of bioriented polar chromosome:

“What determines the direction of movement for polar chromosomes, causing them to migrate toward the equator rather than the spindle pole? Building on our proposed mechanism for chromosome centering during metaphase⁹⁶, and a theoretical model of prometaphase congression⁹⁷, we suggest that this direction is set by an asymmetry in the poleward flux of kinetochore microtubules. Specifically, kinetochore microtubules oriented toward the distant pole are expected to exhibit faster poleward flux than those facing the nearer pole, thereby producing a net force that directs chromosome movement toward the spindle center. Future research will clarify the relevance of this model to polar chromosome congression and identify the factors that control their directional movement.”

The Reviewer: 2) It would be better if the authors could address how CENP-E opposes the kinase activity of centrosomal Aurora A and centromere/kinetochore-localized Aurora B?

The authors: We now address possible mechanisms in the Discussion.

“Mechanistically, we propose that CENP-E stabilizes nascent end-on kinetochore–microtubule attachments, particularly near the spindle poles, where high Aurora A activity likely enhances Aurora B signaling and promotes microtubule detachment from the kinetochore in the absence of CENP-E activity⁸⁸ (Fig. 8a). This model is supported by *in vitro* data showing that Ndc80 and CENP-E are sufficient for lateral-to-end-on transitions⁸⁹. Additionally, it has been shown that perturbations of CENP-E reduce the number of microtubules in kinetochore fibers of aligned chromosomes at the metaphase plate^{85,90}. We suggest that kinetochores that are unable to form stable end-on attachment in the absence of CENP-E or its activity during early prometaphase are likely driven poleward via corona-associated dynein²⁸ and the flux of microtubules laterally attached to chromosomes⁴³.”

“The notion that CENP-E is involved in end-on conversion has been proposed earlier, in particular that CENP-E maintains lateral attachments¹¹. However, our findings suggest that CENP-E may directly promote end-on conversion and biorientation independently of gliding or stabilizing lateral attachments. Different roles of CENP-E independent of gliding were proposed previously^{21,94,95}, but their contribution to chromosome congression, **specifically in stabilizing end-on attachments**, remains to be tested.”

3) It would be better if the authors could test whether distinct pools of CENP-E on the kinetochore and fibrous corona promote the end-on attachment and drive the movement of polar chromosomes, respectively?

The authors: We now clarify this point in the Discussion. In addition to the new text copied above under point 2, we added the following:

“In addition to phosphorylating KMN network components, Aurora kinases inhibit the stabilization of end-on attachments by promoting the expansion of the fibrous corona on unattached chromosomes (Fig. 8b). Previously, it was proposed that competition between corona, which can nucleate and sort microtubules^{33,49} or serve as a hub for capturing microtubules⁵², and

Ndc80 complex regulates microtubule engagement and end-on conversion⁵⁰. This model is supported by our own and previous⁴⁸ observations that the corona is almost completely removed upon acute Aurora B inhibition, which we show triggers immediate congression. Furthermore, we show that CENP-E and Astrin exhibit binary occupancy at kinetochores, where each kinetochore is predominantly enriched for either CENP-E or Astrin, but rarely both, suggesting a switch-like mechanism that reflects the formation of end-on attachments. Finally, in cells expressing a constitutively dephosphorylated Hec1, polar chromosomes congress only when CENP-E is fully depleted, but not when CENP-E is inhibited⁸⁸, which we show leads to a hyperexpanded corona.”

“Recent studies have shown that CENP-E occupies distinct pools on the kinetochore and fibrous corona³⁰, with BubR1 and the RZZ complex recruiting these pools, respectively^{30,32,80}. Since we have shown that congression initiation completely depends on BubR1, we propose this process is primarily driven by the kinetochore-localized CENP-E.”

4) If I understand correctly, in the working model showing in Figure 8B, the blunt line from Aurora B to Knl1/Ndc80 should be to the arrow line between Knl1/Ndc80 and End-on attachment.

The authors: We have now changed Figure caption to explain better the meaning of blunt lines.

“Lines with arrows indicate activation, and blunt lines indicate deactivation by phosphorylation or direct physical inhibition.”

5) In lines 376-378, the authors mentioned that “On the other hand, inhibition and co-inhibition of Bub1 (Budding uninhibited by benzimidazoles or Haspin, factors essential for the recruitment of Aurora B to the outer and inner centromere, respectively 61–64”. The following literatures should be cited as well: Wang F et al., Science 2010 (PMID: 20705812); Yamagishi Y et al., Science 2020 (PMID: 20929775); Liang C et al., J Cell Biol 2020 (PMID: 31868888).

The authors: We thank the reviewer for mentioning these important papers. We have cited these papers at the place where the reviewer has suggested.

“On the other hand, inhibition and co-inhibition of Bub1 (Budding uninhibited by benzimidazoles 1) or Haspin, factors essential for the recruitment of Aurora B to the outer and inner centromere, respectively⁶⁶⁻⁷¹, did not induce chromosome congression in the absence of CENP-E activity, as well as inhibition of Polo-like kinase 1 (Plk1) kinase, and depletion of Bub1-related (BubR1) pseudokinase (Fig. 5c; and Extended Data Fig. 5n).”

6) In lines 382-384, the authors mentioned that “the control group was Monopolar spindle 1 (Mps1) kinase (Fig. 5C; and Fig. S4N). This implies that centrosome localized Aurora A and the outer-kinetochore localized Aurora B are major factors that limit the initiation of congression in the absence of CENP-E”. In addition, in lines 574-576, the authors mentioned that “Polar chromosomes initially exhibit lateral attachments of kinetochores to microtubules, which fail to quickly convert to stable end-on attachments without CENP-E due to the high activity of outer-

kinetochore localized Aurora B (AurB)". However, the authors did not provide evidence showing that it is the outer-kinetochore localized Aurora B that executes the function.

The authors: We thank the reviewer for this comment and we completely agree. We have now rewritten this part of the manuscript to emphasize that our results only suggest that Aurora B pool at the kinetochore (not specifically on the outer kinetochore) is important for limiting the initiation of congression in the absence of CENP-E. We have also deleted "outer-kinetochore" from the Figure caption.

"Altogether, these results suggest that centrosome localized Aurora A and the kinetochore localized Aurora B are limiting the initiation of congression in the absence of CENP-E."

"Chromosome congression in the absence of CENP-E is primarily limited by the kinetochore-localized Aurora B pool."

Reviewer #3 (Remarks to the Author):

Bioriented kinetochore-microtubule attachments are essential for the accurate segregation of chromosomes. The authors explore how CENP-E initiates congression and promotes end-on attachments to ensure proper biorientation. In previous studies, it has been shown that CENP-E is essential for end-on attachments. Here using a combination of gentle high-resolution live-cell microscopy and CENP-E inhibitor treatment the authors show that the kinesin is required to form or stabilize end-on attachments rather than directly powering the chromosome movement. This is likely because CENP-E accelerates congression initiation by opposing Aurora kinase activity, but chromosome movement toward the equator does not require CENP-E. Also stabilization of end-on attachments is delayed in the absence of CENP-E due to Aurora kinase-mediated phosphorylation of microtubule-binding proteins on kinetochores and the expansion of the fibrous corona. The authors speculate that CENP-E reduces Aurora B-mediated phosphorylation of outer kinetochore proteins in a BubR1-dependent manner and thereby stabilizes end-on attachments which is necessary for the full removal of the fibrous corona and biorientation.

Major comments.

The reviewer: Figure 2 shows change in Mad2 levels associated with lateral to end-on conversion. Can the authors compare levels in poleward vs non-poleward movement of sisters to indicate whether one of the sisters is more end-on tethered than the other? This is important because the conclusions are currently made using one marker Mad2 which has dynein-dependent and independent pathways to leave the kinetochore.

Kuhn and Dumont 2017 use SKAP as a marker to distinguish end-on attachment from lateral attachment based long-lived force. Having a second marker beyond Mad2 can help some of the key conclusions.

The authors: Regarding the analysis of poleward versus non-poleward movements in Mad2-expressing cells, we have opted not to pursue this due to the relatively long intervals between consecutive imaging frames, which is insufficient to capture the brief, transient poleward kinetochore movements that occur during congression. These longer intervals were necessary to image the entire mitotic spindle in the z-dimension in cells expressing both CENP-A and Mad2.

We are especially grateful for the reviewer's suggestion regarding the use of additional markers for end-on attachment. To more comprehensively assess end-on attachment status on congressing kinetochores in a high-throughput manner, we conducted a large-scale immunofluorescence-based assay under four experimental conditions (see also comment 1 by reviewer 3):

- 1) Continuous CENP-E inhibitor treatment
- 2) Cells fixed 8 minutes after inhibitor washout
- 3) Cells fixed 15 minutes post-washout
- 4) DMSO-treated control cells

We stained for the Astrin/SKAP complex, a well-established marker of stable end-on attachments, and for CENP-E, a fibrous corona marker. For each condition, we quantified the mean intensity of Astrin and CENP-E at every polar kinetochore, as well as at randomly selected aligned kinetochores. We also measured the distance of each kinetochore from the nearest spindle pole to spatially contextualize these signals.

“To independently test whether stabilization and maturation of end-on microtubule attachments occur at congressing chromosomes, we stained RPE-1 cells for Astrin, a positive marker of stable end-on attachments^{47,48} and CENP-E, a marker of fibrous corona^{30,49}. To capture the full range of kinetochore positions from the spindle poles to the equatorial plate, we analyzed cells under CENP-E inhibition, at two time points following CENP-E reactivation, and in DMSO-treated prometaphase cells undergoing congression (Fig. 3f). We quantified the intensity of Astrin and CENP-E at all polar kinetochores and at randomly selected aligned kinetochores, together with their distance from the nearest spindle pole. Following CENP-E reactivation, Astrin levels at polar kinetochores progressively increased as kinetochores were found closer to equatorial plane, reaching values comparable to those at aligned kinetochores (Fig. 3g). Increase in Astrin on congressing kinetochores mirrored both Mad2 loss (Fig. 2f) and end-on attachment maturation observed by STED microscopy (Fig. 3d) as the distance from the pole increased. Notably, by 15 minutes post-washout, Astrin levels at congressing kinetochores became indistinguishable from those in DMSO-treated prometaphase cells (Fig. 3g). In contrast, CENP-E levels declined as kinetochores moved toward the metaphase plate across conditions (Fig. 3h). CENP-E and Astrin display a mutually exclusive localization pattern at kinetochores: kinetochores enriched in CENP-E exhibit low Astrin levels, whereas those with high Astrin show reduced CENP-E presence (Fig. 3i). These results support a model in which chromosome congression involves the progressive capture and stabilization of end-on microtubule attachments (Fig. 3j), both following CENP-E reactivation and during unperturbed mitosis. Overall, congressing chromosomes are biochemically characterized by intermediate levels of checkpoint proteins (e.g., Mad2), intermediate Astrin levels indicating the maturation of end-on attachments, and reduced levels of fibrous corona proteins like CENP-E, which reflects ongoing corona stripping.”

The reviewer: Lateral microtubule attachments were found in EM studies of congressed kinetochores (Magidson et al., 2011). Here, using STED the authors show that congressing chromosomes are bioriented a few microns away from centrosomes. The former was an observation made in unperturbed cells while this study focusses on chromosomes positioned near the poles. The authors may benefit discussing these two cases, instead of ruling out one or the other. In fact, the authors do find that Aurora kinases (at poles) can delay congression in the absence of CENP-E activity. Moreover, it will help explain why only some but not all kinetochores remain uncongressed in CENP-E inhibitor treated cells.

The authors: We thank the reviewer for this important comment. We agree that this point warrants a more detailed discussion, but we disagree that our model is ruling out observations from the paper the reviewer is mentioning. As the reviewer noted, during early prometaphase, shortly after NEBD, somatic cells pass through a roseate stage in which lateral kinetochore–microtubule attachments dominate, as elegantly demonstrated by EM studies from the Khodjakov lab (Magidson et al., 2011). However, more recent work from the same lab (doi: 10.1016/j.cub.2022.01.013) showed that, shortly after this stage, the majority of chromosomes in intact cells are already bioriented. This indicates that in intact RPE-1 cells, biorientation and the establishment of end-on attachments occur rapidly.

In the accompanying manuscript (10.1101/2024.09.29.614573v1), we argue that for approximately 20% of chromosomes, CENP-E is essential to overcome the high activity of Aurora kinases near the spindle poles, which otherwise destabilize end-on kinetochore–microtubule

attachments. We further show that such centrosome-kinetochore feedback explains why only a subset of chromosomes require CENP-E activity: these chromosomes are positioned in regions of elevated Aurora A activity. This activity, either directly or via upregulation of Aurora B at kinetochores, contributes to the destabilization of end-on attachments on polar chromosomes, thus necessitating CENP-E for their rapid congression. We now discuss these notions in more detail in this manuscript:

“Our model predicts that chromosome congression initiates without CENP-E activity when the KMN network has high microtubule affinity, stabilizing nascent end-on attachments and leading to gradual fibrous corona stripping We have demonstrated this process on polar chromosomes following acute Aurora kinase inhibition in the absence of CENP-E. Additionally, we have shown that this process also occurs during intact mitosis at kinetochores distant from the centrosome, where end-on attachments stabilize independently of CENP-E⁸⁸. In the presence of CENP-E, we propose that this process occurs at all kinetochores during early prometaphase, explaining the rapid biorientation and congression observed in healthy cells.”

“We do not exclude the possibility that CENP-E can facilitate chromosome movement if very close to the pole by gliding polar kinetochores along highly detyrosinated microtubules away from the Aurora A gradient^{4,27}. However, polar chromosomes rarely encounter a high Aurora A gradient during intact mitosis in healthy cells, as they do not approach the centrosomes^{33,37}. This is in stark contrast to cancer cells, where a small number of polar chromosomes move closer to the centrosomes early in mitosis, leading to congression delays⁸. We propose that in healthy cells, the avoidance of regions near the centrosome by chromosomes may play a pivotal role in enabling rapid biorientation and congression. In the accompanying manuscript⁸⁸, we provide evidence that centrosomes through Aurora A deliver a crucial spatial signal that determines the need for CENP-E during chromosome congression, as CENP-E counteracts the combined activity of Aurora kinases to initiate chromosome movement.“

“Mechanistically, we propose that CENP-E stabilizes nascent end-on kinetochore–microtubule attachments, particularly near the spindle poles, where high Aurora A activity likely enhances Aurora B signaling and promotes microtubule detachment from the kinetochore in the absence of CENP-E activity⁸⁸ (Fig. 8a). This model is supported by *in vitro* data showing that Ndc80 and CENP-E are sufficient for lateral-to-end-on transitions⁸⁹. Additionally, it has been shown that perturbations of CENP-E reduce the number of microtubules in kinetochore fibers of aligned chromosomes at the metaphase plate^{85,90}. We suggest that kinetochores that are unable to form stable end-on attachment in the absence of CENP-E or its activity during early prometaphase are likely driven poleward via corona-associated dynein²⁸ and the flux of microtubules laterally attached to chromosomes⁴³.”

The reviewer: Would it help the authors to exploit STLC or monastrol release for studying biorientation of kinetochores that are randomly near the equator (far away from poles) but attempting to align at kinetochores? Or can they compare congression efforts (persistent travel or Inter-kinetochore distances) close to the equator versus pole using the data they have?

The authors: We agree that this is a great idea. In accompanying manuscript (10.1101/2024.09.29.614573v1), we demonstrated that the distance of a chromosome from the centrosome correlates with its likelihood of initiating congression in the absence of CENP-E activity. This correlation is markedly weaker following acute reactivation of CENP-E, suggesting

that CENP-E is particularly important for initiating congression of chromosomes located near the spindle poles.

In the accompanying manuscript we have also shown that acute monastrol treatment in cells pre-treated with the CENP-E inhibitor, did not lead to increased initiation of congression compared with cells treated with the CENP-E inhibitor alone. Additionally, when monastrol washout was performed prior to acute CENP-E inhibition, the number of chromosomes requiring CENP-E for rapid congression remained comparable to that observed in cells that had entered mitosis in the continuous presence of the inhibitor. Together, these findings argue that CENP-E dependence is spatially defined by the proximity of chromosomes to the spindle poles, rather than by their relative spatial organization within the spindle at a given moment. The consistent requirement for CENP-E across these different conditions supports the conclusion that its role is dictated by pole proximity rather than by the broader spindle architecture.

The reviewer: Minor comments?

Figure 1A – why does the 1.5mm line span the magnified crop with a 5micron scale bar?

The authors: This is corrected.

The reviewer: Please include immunoblot or immunofluorescence to show extent of CENP-E depletion.

The authors: Immunofluorescent images of non-targeting controls and depleted cells, together with analysis of depletion of CENP-E, as well as other proteins depleted in the study, compared to untreated controls, was presented in Supplementary Fig. 1, now Extended Data Fig. 2. Immunofluorescent images of Hec1 after depletions of the protein by various siRNAs together with quantification is presented in Fig. 7.

The reviewer: Figure S3A and B: typo ‘reactivated’

The authors: This is corrected.

The reviewer: In figure 6E – is there a lowering of intensity with time?

The authors: Yes. This was probably caused by larger photobleaching in this cell, as this experiment was conducted on an older microscopy setup. However, we reason that the fact that most chromosomes returned to the metaphase plate in this cell, suggests that photobleaching was not accompanied with phototoxicity that could influence kinetochore movements.

The reviewer: Figure 7 interpretation is somewhat confusing. In HEC1-depleted cells end-on attachments cannot form, so we do expect that CENP-E;HEC1 codepletion to present severe misalignment. Also HEC1 depletion causes a prolonged mitotic arrest, and during this arrest kinetochores often worsen their congression status making it difficult to build a direct correlation between congression status and HEC1 levels. Live-cell imaging and then a fixation at the end of the tracking may be helpful here for clear conclusions.

The authors: We completely agree with the reviewer that this is an obvious limit of this experiment. However, due to the already complex experimental setup, performing this experiment

would take much longer time than a reasonable revision. Thus, we have decided to omit this experiment. We have now included a sentence about the limit of this experiment in the Result section:

“We note that extensive misalignment in cells with minimal Hec1 could be exacerbated by mitotic prolongation.”

The reviewer: Please clarify these sentences:

“This suggests that issues with error correction, caused by Aurora B inhibition⁵⁰, limit the complete congression of some polar chromosomes.”

In figure legend 8 “Functional negative feedback loop of signaling interactions”

The authors: Both sentences are now clarified in the revised version.

1) “). This suggests that incomplete congression of polar chromosomes may result from error correction defects, such as unrepaired merotelic attachments, a known effect of Aurora B inhibition⁵⁵.”

2) “(b) A negative feedback loop involving Aurora B, CENP-E, BubR1–PP2A, the fibrous corona, and outer kinetochore proteins (Knl1 and Hec1/Ndc80), which self-limits by promoting end-on attachment formation and initiating chromosome congression, thereby reducing the upstream signals that sustain the loop.”

Rebuttal letter Vukušić&Tolić, NCOMMS-24-76214A

We thank the reviewers for highlighting parts that need to be clarified. Below, we provide detailed responses to each comment. All modifications to the manuscript are highlighted in blue for clarity.

REVIEWER COMMENTS

Reviewer #1 (Remarks to the Author):

Thank you for addressing all my concerns. The manuscript has been greatly improved. I have no further concerns.

Reviewer #2 (Remarks to the Author):

The authors have addressed all my comments. I therefore support the publication of this important study.

The authors: We thank Reviewers 1 and 2 for assessing the revised manuscript:

Reviewer #3 (Remarks to the Author):

Reviewer #3 (Remarks to the Author):

In the manuscript titled “CENP-E initiates chromosome congression by opposing Aurora kinases to promote end-on attachments”. The authors have addressed all my queries satisfactorily. Recommendations below are largely clarifications on literature (citations) and data analysis.

The authors: We are grateful to the reviewer for the thorough assessment of our revised manuscript:

1) In figure 3J, ‘congression’ and ‘alignment’ are somewhat confusing as it seems like aligned chromosomes have to be at equator whereas congressed is based on movement. Would it help them to call it ‘congressing’ and ‘aligned’ based on the precise phenotype? Or explain in results text how these phenotypes were separated in time.

The authors: We thank the reviewer for raising this point. In Fig. 3j, we have replaced ‘aligned’ with ‘fully aligned’ to make clear that this refers to complete alignment of the chromosomes at the metaphase plate, as illustrated in the figure:

2) The above comment is important in the context of abstract statement “These findings support a unified 20 model of chromosome movement in which congression is intrinsically coupled to biorientation.” Does the statement indicated that alignment is not coupled to biorientation, and only congression is coupled to biorientation? Some text clarification or abstract revision would be helpful for clarity.

The authors: We have clarified this in the text in the Discussion section:

“The relationship between chromosome congression and biorientation has remained unclear. It has been established that maintenance of chromosome position at the equator after congression requires stable kinetochore–microtubule attachments and biorientation^{1,13}. In contrast, congression

was thought to occur via two distinct pathways, either through biorientation or through CENP-E-mediated mechanisms^{1,4,13,19}.”

3) Citation: Original papers missed.

3.1 Consider discussing your findings in the context of CENPE siRNA and inhibitor work from PMC6080938

The authors: We thank the reviewer for pointing out this paper. We have now cited it and added a brief discussion on the relevance of the model presented in that and subsequent papers in relation to the model proposed in this manuscript:

“While CENP-E has been proposed to maintain lateral attachments¹¹, our findings suggest it may also directly promote end-on conversion and biorientation, independently of its gliding activity. Although non-gliding roles for CENP-E have been proposed^{21,91,92}, their contribution to stabilizing end-on attachments during congression remains to be tested. For example, CENP-E-mediated transport of laterally attached microtubules along the kinetochore surface that delivers their plus-ends near kinetochores^{33,93}, could facilitate KMN network function and stabilize end-on attachments.”

3.2 Revise references related to “expressing the SAC protein Mad2, which accumulates on kinetochores lacking stable end-on attachments^{44–46}”. See figure 4 of PMID: 26258631 on correlative LM/EM status for Mad2.

The authors: We fully agree with the reviewer and appreciate the suggestion. We have included the proposed reference in the revised manuscript:

3.3 Revise references related to “Astrin, a positive marker of stable end-on attachments^{47,48}” : Astrin-SKAP was established as proteins at the kinetochore in the papers cited (Ref 47 and 48) and the Astrin-SKAP complex was demonstrated as an end-on attachment marker in PMC3748344 and PMC5461026.

The authors: We agree with the reviewer and are grateful for this suggestion. We have included the proposed references in the revised manuscript:

4) In figure 8 model, if CENPE initiates end-on attachments, how are bioriented kinetochores seen in CENPEinhibitor treated cells. Should it be ‘CENPE initiates end-on attachments of polar chromosomes’? Some specification in legend and cartoon would help clarity.

The authors: We thank the reviewer for raising this point. As noted in the legend of Figure 8, we have already indicated that this mechanism is relevant for polar chromosomes, whereas chromosomes further from the pole undergo biorientation independently of CENP-E, though with fewer microtubules in k-fibers, as also discussed in the Discussion section:

“(a) Aurora kinases inhibit congression initiation by phosphorylating the Ndc80 tail near centrosomes (1). On polar kinetochores, CENP-E-BubR1 facilitates early end-on attachment formation near the Aurora A gradient (2), triggering a decline in Aurora B activity, loss of Mad2

from the kinetochore, and stabilization of Ndc80–microtubule binding, all preceding fast kinetochore movement (3).”